# Attenuated fusogenicity and pathogenicity of SARS-CoV-2 Omicron variant

Rigel Suzuki[1,27], Daichi Yamasoba[2,3,27], Izumi Kimura[2,27], Lei Wang[4,5,27], Mai Kishimoto[6,27], Jumpei Ito[2,27], Yuhei Morioka[1], Naganori Nao[7,8], Hesham Nasser[9,10], Keiya Uriu[2,11], Yusuke Kosugi[2,12,13], Masumi Tsuda[4,5], Yasuko Orba[6,14], Michihito Sasaki[6,14], Ryo Shimizu[9], Ryoko Kawabata[15], Kumiko Yoshimatsu[16], Hiroyuki Asakura[17], Mami Nagashima[17], Kenji Sadamasu[17], Kazuhisa Yoshimura[17], The Genotype to Phenotype Japan (G2P-Japan) Consortium*, Hirofumi Sawa[6,7,8,14], Terumasa Ikeda[9], Takashi Irie[15], Keita Matsuno[8,14,18 ✉], Shinya Tanaka[4,5 ✉], Takasuke Fukuhara[1 ✉] & Kei Sato[2,11,19 ✉]

The emergence of the Omicron variant of SARS-CoV-2 is an urgent global health concern[1]. In this study, our statistical modelling suggests that Omicron has spread more rapidly than the Delta variant in several countries including South Africa. Cell culture experiments showed Omicron to be less fusogenic than Delta and than an ancestral strain of SARS-CoV-2. Although the spike (S) protein of Delta is efficiently cleaved into two subunits, which facilitates cell–cell fusion[2,3], the Omicron S protein was less efficiently cleaved compared to the S proteins of Delta and ancestral SARS-CoV-2. Furthermore, in a hamster model, Omicron showed decreased lung infectivity and was less pathogenic compared to Delta and ancestral SARS-CoV-2. Our multiscale investigations reveal the virological characteristics of Omicron, including rapid growth in the human population, lower fusogenicity and attenuated pathogenicity.

Newly emerging SARS-CoV-2 variants need to be carefully monitored for potentially increased transmissibility, pathogenicity and resistance to vaccine-induced immunity and antiviral drugs. As of December 2021, the World Health Organization (WHO) has defined five variants of concern (VOCs)—Alpha (B.1.1.7), Beta (B.1.351), Gamma (P.1), Delta (B.1.617.2 and AY lineages) and Omicron (originally B.1.1.529, then reclassified into BA lineages)—as well as two variants of interest, Lambda (C.37) and Mu (B.1.621)[4]. These SARS-CoV-2 variants pose an ongoing threat to human society. For example, the Alpha variant, which has an N501Y substitution in its S protein, transmits more efficiently than ancestral SARS-CoV-2[5]; and the Beta, Gamma and Mu variants, which bear the E484K substitution, exhibit robust resistance to neutralizing antibodies that are elicited by vaccination and natural SARS-CoV-2 infection[6–13]. In addition, we have previously shown that the Delta variant is more highly pathogenic than the D614G-bearing early-pandemic virus in a hamster model[2].

In January 2022, the Omicron variant (originally B.1.1.529 lineage) represents the most recently recognized VOC[4]. The variant was first detected in South Africa on 24 October 2021 (GISAID ID: EPI_ISL_7605742). On 24 November 2021, the B.1.1.529 lineage, a descendant of the SARS-CoV-2 B.1.1 lineage[14], was reported to WHO as a novel variant spreading in South Africa[15]. On 25 November 2021, this variant was identified as concerning as a result of its potential to outcompete the Delta variant in Gauteng province, South Africa[16,17]. Because of the potential risk that this newly emerged variant posed to global health, WHO rapidly classified B.1.1.529 as a VOC and designated it the Omicron variant on 26 November 2021 (ref. [1]).

Omicron seems to be spreading rapidly, especially relative to the spread rate of Delta, which was the predominant variant worldwide in December 2021. The virological features of Omicron, such as its pathogenicity and its resistance to antiviral immunity and drugs, are unclear. Compared to the original SARS-CoV-2 strain (B lineage, strain Wuhan-Hu-1, GenBank accession no. NC_045512.2)[18], Delta (for example, B.1.617.2 lineage, strain TKYTK1734, GISAID ID: EPI_ISL_2378732) has 45 nucleotide mutations across its genome, including 8 nonsynonymous or insertion and deletion (indel) mutations in its S protein. By contrast, Omicron (for example, BA.1 lineage, strain TY38-873, GISAID ID: EPI_ISL_7418017) contains 97 nucleotide mutations across its genome, including 33 nonsynonymous or indel mutations in its S protein (Supplementary Table 1). The higher number of mutations in Omicron—and

[1]Department of Microbiology and Immunology, Graduate School of Medicine, Hokkaido University, Sapporo, Japan. [2]Division of Systems Virology, Department of Infectious Disease Control, International Research Center for Infectious Diseases, The Institute of Medical Science, The University of Tokyo, Tokyo, Japan. [3]Faculty of Medicine, Kobe University, Kobe, Japan. [4]Department of Cancer Pathology, Faculty of Medicine, Hokkaido University, Sapporo, Japan. [5]Institute for Chemical Reaction Design and Discovery (WPI-ICReDD), Hokkaido University, Sapporo, Japan. [6]Division of Molecular Pathobiology, International Institute for Zoonosis Control, Hokkaido University, Sapporo, Japan. [7]Division of International Research Promotion, International Institute for Zoonosis Control, Hokkaido University, Sapporo, Japan. [8]One Health Research Center, Hokkaido University, Sapporo, Japan. [9]Division of Molecular Virology and Genetics, Joint Research Center for Human Retrovirus infection, Kumamoto University, Kumamoto, Japan. [10]Department of Clinical Pathology, Faculty of Medicine, Suez Canal University, Ismailia, Egypt. [11]Graduate School of Medicine, The University of Tokyo, Tokyo, Japan. [12]Laboratory of Systems Virology, Institute for Frontier Life and Medical Sciences, Kyoto University, Kyoto, Japan. [13]Graduate School of Pharmaceutical Sciences, Kyoto University, Kyoto, Japan. [14]International Collaboration Unit, International Institute for Zoonosis Control, Hokkaido University, Sapporo, Japan. [15]Institute of Biomedical and Health Sciences, Hiroshima University, Hiroshima, Japan. [16]Institute for Genetic Medicine, Hokkaido University, Sapporo, Japan. [17]Tokyo Metropolitan Institute of Public Health, Tokyo, Japan. [18]Division of Risk Analysis and Management, International Institute for Zoonosis Control, Hokkaido University, Sapporo, Japan. [19]CREST, Japan Science and Technology Agency, Saitama, Japan. [27]These authors contributed equally: Rigel Suzuki, Daichi Yamasoba, Izumi Kimura, Lei Wang, Mai Kishimoto, Jumpei Ito. *A list of authors and their affiliations appears at the end of the paper. ✉e-mail: matsuk@czc.hokudai.ac.jp; tanaka@med.hokudai.ac.jp; fukut@pop.med.hokudai.ac.jp; KeiSato@g.ecc.u-tokyo.ac.jp

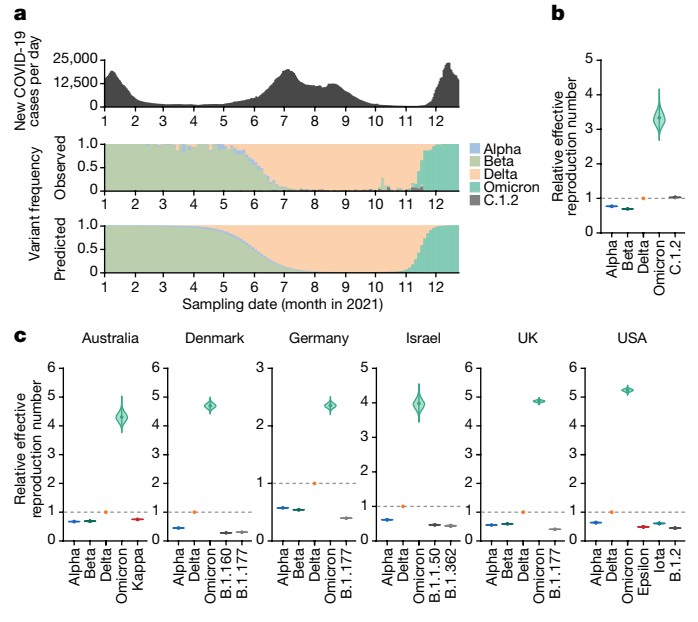

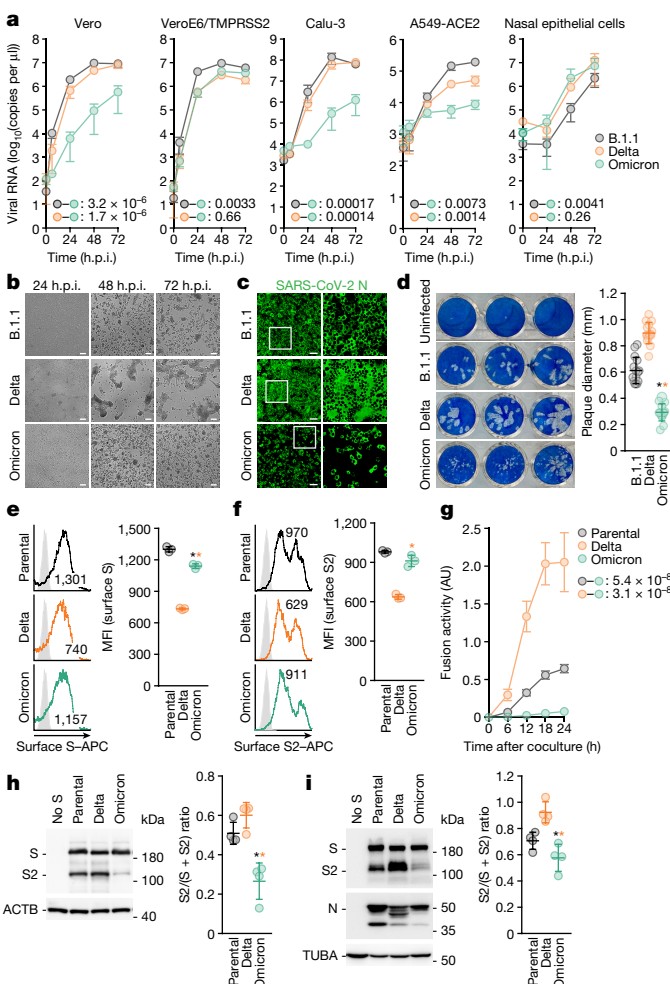

**Fig. 1 | Epidemic dynamics of Omicron. a**, Top, the seven-day average of new COVID-19 cases reported per day. Middle, the frequency of the top five viral lineages in the sequenced samples. Bottom, the frequency of the top five viral lineages predicted by our Bayesian statistical model. The data are from South Africa, from 1 January 2021 to 24 December 2021. The lineage frequency (middle and bottom) is summarized in three-day bins. The frequencies of all viral lineages are shown in Extended Data Fig. 1. **b**, **c**, Estimation of the relative effective reproduction number of each viral lineage, assuming a fixed generation time of 5.5 days. Values are shown relative to Delta (the Delta value is set at 1) in South Africa (**b**) and other six countries (Australia, Denmark, Germany, Israel, the UK and the USA) (**c**). The posterior distribution (violin), posterior mean (dot) and 95% credible interval (bar) are indicated.

**Fig. 2 | Virological features of Omicron in vitro. a**, Growth kinetics of Omicron. B.1.1 virus, Delta and Omicron were inoculated into cells, and the copy number of the viral RNA in the supernatant was quantified by quantitative PCR with reverse transcription (RT–qPCR). **b**, Bright-field images of infected VeroE6/TMPRSS2 cells (multiplicity of infection (m.o.i.) of 0.01). **c**, Immunofluorescence staining. Infected VeroE6/TMPRSS2 cells (m.o.i. = 0.01) at 24 h.p.i. were stained with anti-SARS-CoV-2 N antibody. Higher-magnification views of the regions indicated by squares are shown on the right. Scale bars, 100 μm (**b**, **c**). **d**, Plaque assay. Left, representative figures. Right, summary of the diameter of plaques (15 plaques per virus). **e**, **f**, Expression of the S protein on the cell surface. Left, representative histogram stained with anti-S1/S2 polyclonal antibody (**e**) or anti-S2 monoclonal antibody (**f**). The number in the histogram indicates the mean fluorescence intensity (MFI). Grey histograms indicate isotype controls. Right, summary of the surface S MFI. **g**, SARS-CoV-2 S-based fusion assay. The fusion activity was measured as described in the Methods, and fusion activity (arbitrary units; AU) is shown. **h**, **i**, Left, representative western blots of S-expressing cells (**h**) or SARS-CoV-2-infected VeroE6/TMPRSS2 cells (m.o.i. = 0.01) at 48 h.p.i. (**i**). ACTB (**h**) or TUBA (**i**) are internal controls. Right, the ratio of S2 to the full-length S plus S2 proteins. Data are mean ± s.d. (**a**, **d**–**i**). Assays were performed in quadruplicate (**a**, **g**–**i**) or triplicate (**e**–**f**). Each dot indicates the result from an individual plaque (**d**) and an individual replicate (**e**, **f**, **h**, **i**). Statistically significant differences versus B.1.1 and Delta through time points were determined by multiple regression (**a**, **g**). Familywise error rates (FWERs) calculated using the Holm method are indicated. Statistically significant differences (*$P < 0.05$) versus B.1.1 and Delta were determined by two-sided Mann–Whitney $U$-test (**d**) or by two-sided paired Student's $t$-test (**e**, **f**, **h**, **i**) without adjustment for multiple comparisons.

particularly those in the S protein—may affect the viral phenotype. Here we investigate the virological characteristics of Omicron in human cells in vitro and hamsters.

## Epidemic dynamics of Omicron

In South Africa, both the number of cases of COVID-19 and the frequency of the Omicron variant increased rapidly in November 2021 (Fig. 1a, Extended Data Fig. 1). To estimate the relative effective reproduction numbers of SARS-CoV-2 lineages including Omicron in South Africa, we constructed a Bayesian statistical model that represents the dynamics of viral lineage frequency[19–21]. Our statistical analysis showed that the effective reproduction number of Omicron in South Africa was 3.31-fold higher than that of Delta (95% credible interval: 2.95–3.72; Fig. 1b). Our results are consistent with a recent study[22]. In addition, similar to the results in South Africa (Fig. 1b), the effective reproduction numbers of Omicron were greater than those of Delta in the six other countries in which more than 1,500 Omicron sequences had been reported (Australia, Denmark, Germany, Israel, the UK and the USA) (Fig. 1c). As of 7 January 2022, more than 200,000 Omicron sequences had been reported in approximately 100 countries. These results suggest that Omicron has spread extremely rapidly and may outcompete Delta around the world in the near future.

## Virological features of Omicron in vitro

To elucidate the virological characteristics of Omicron, we obtained an Omicron isolate (strain TY38-873). A D614G-bearing early-pandemic B.1.1 isolate (strain TKYE610670)[2] and a Delta isolate (B.1.617.2 lineage, strain

TKYTK1734)[2] were used as controls. Although the growth of Omicron in VeroE6/TMPRSS2 and primary human nasal epithelial cells was comparable to that of Delta, Omicron was less replicative than Delta in Vero,

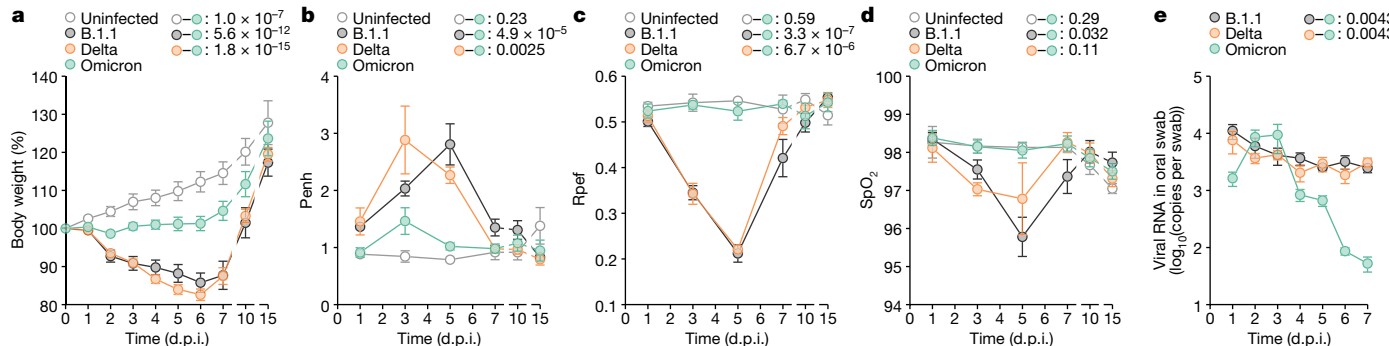

**Fig. 3 | Time-course dynamics of Omicron in vivo.** Syrian hamsters were intranasally inoculated with saline ($n = 6$, uninfected control), B.1.1 ($n = 6$), Delta ($n = 6$) or Omicron ($n = 6$). Six hamsters of the same age were mock-infected. Body weight (**a**), Penh (**b**), Rpef (**c**), SpO$_2$ (**d**) and viral RNA load in oral swabs (**e**) were routinely measured. Data are mean ± s.e.m. In **a**–**d**, statistically significant differences versus B.1.1 and Delta through time points were determined by multiple regression. In **e**, statistically significant differences of the dynamics versus B.1.1 and Delta were determined by a permutation test. FWERs calculated using the Holm method are indicated.

Calu-3, A549-ACE2 and HeLa-ACE2/TMPRSS2 cells (Fig. 2a, Extended Data Fig. 2). Omicron and the other isolates replicated in A549-ACE2 cells but did not in A549 cells (Fig. 2a, Extended Data Fig. 2), suggesting that Omicron uses the ACE2 molecule as the receptor for infection. Although the growth kinetics of Omicron and Delta in VeroE6/TMPRSS2 cells were comparable (Fig. 2a, Extended Data Fig. 2), the morphology of infected cells was quite different: Delta formed larger syncytia than the B.1.1 virus, which is consistent with our previous work[2], whereas Omicron only weakly formed syncytia (Fig. 2b). Immunofluorescence assays at 24 h post-infection (h.p.i.) further showed that VeroE6/TMPRSS2 cells that were infected with Delta exhibited larger multinuclear syncytia than B.1.1-infected cells, whereas cells infected with Omicron did not (Fig. 2c). Moreover, the plaque size in VeroE6/TMPRSS2 cells infected with Omicron was significantly smaller than that in cells infected with Delta (3.06-fold) or the B.1.1 virus (2.08-fold) (Fig. 2d). These data suggest that Omicron is less fusogenic than Delta and an early-pandemic SARS-CoV-2.

To directly assess the fusogenicity of the S proteins of these variants, we performed a cell-based fusion assay[2,23]. The expression level of Omicron S on the cell surface was lower than (when stained with an anti-S polyclonal antibody; Fig. 2e) or comparable to (when stained with an anti-S2 monoclonal antibody; Fig. 2f) that of the D614G-bearing parental S, and Omicron S was more highly expressed on the cell surface than Delta S (Fig. 2e, f). Nevertheless, our fusion assay showed that Omicron S is significantly less fusogenic than Delta S and the parental D614G S (Fig. 2g, Extended Data Fig. 3a). In addition, coculturing S-expressing cells with HEK293-ACE2/TMPRSS2 cells showed that Omicron S only induced multinuclear syncytia at a low level (Extended Data Fig. 3b).

Because Delta infection forms larger syncytia and Delta S exhibits higher fusogenicity with efficient cleavage between S1 and S2 (hereafter, S1/S2 cleavage)[2,3], we hypothesized that the poor syncytium formation and lower fusogenicity of Omicron might be attributable to a low efficacy of S cleavage. Consistent with our previous studies[2,3], in the S-expressing cells, the level of the cleaved S2 subunit was higher for Delta S than for the D614G-bearing parental S (Fig. 2h). In sharp contrast, the level of cleaved S2 of Omicron S was significantly lower than that of Delta S (2.5-fold) and parental S (2.2-fold) (Fig. 2h). Similarly, enhanced S1/S2 cleavage was observed in Delta-infected VeroE6/TMPRSS2 cells, whereas S cleavage was attenuated in Omicron-infected cells (Fig. 2i). Overall, our data suggest that Omicron S is less efficiently cleaved and less fusogenic than the S proteins of Delta and early-pandemic SARS-CoV-2.

## Virological features of Omicron in vivo

To investigate the dynamics of viral replication in vivo and pathogenicity of Omicron, we conducted hamster infection experiments using B.1.1, Delta and Omicron strains. Consistent with our previous study[2],

hamsters that were infected with B.1.1 and Delta exhibited decreased body weight from 2 days post-infection (d.p.i.) (Fig. 3a). Although the body weight of Omicron-infected hamsters was significantly lower than that of uninfected hamsters, it remained significantly higher than that of B.1.1-infected and Delta-infected hamsters (Fig. 3a). We then quantitatively analysed the lung function of infected hamsters as reflected by three parameters; namely, enhanced pause (Penh) and the ratio of time to peak expiratory follow relative to the total expiratory time (Rpef), which are surrogate markers for bronchoconstriction or airway obstruction; and subcutaneous oxygen saturation (SpO$_2$). As shown in Fig. 3b–d, the B.1.1-infected and Delta-infected hamsters exhibited respiratory disorders according to these three parameters. By contrast, in Omicron-infected hamsters, the Penh value was significantly lower than that in B.1.1-infected and Delta-infected hamsters (Fig. 3b), and the Rpef value was significantly higher than that in the other two infected groups (Fig. 3c). More specifically, the Rpef and SpO$_2$ values of Omicron-infected hamsters were comparable to those of uninfected hamsters (Fig. 3c, d). These data suggest that Omicron is less pathogenic than the B.1.1 and Delta viruses.

We next assessed viral production by routinely collecting oral swabs from infected hamsters. As shown in Fig. 3e, the dynamics of the viral RNA load in oral swabs from Omicron-infected hamsters were significantly different from those of B.1.1-infected and Delta-infected hamsters. The viral RNA loads of B.1.1 and Delta peaked at 1 d.p.i. and were relatively stable by 1 week (Fig. 3e). In sharp contrast, the viral RNA load of Omicron peaked at 2–3 d.p.i., surpassed those of B.1.1 and Delta transiently at this period and then rapidly decreased (Fig. 3e). A clustering analysis also showed that the dynamics of the viral RNA in oral swabs of Omicron were clearly separated from those of the other two viruses (Extended Data Fig. 4). These data suggest that the dynamics of viral excretion to the oral cavity of Omicron are different from those of B.1.1 and Delta.

To further investigate virus spread in infected hamsters, an immunohistochemistry (IHC) analysis of viral nucleocapsid (N) protein was conducted using samples from the respiratory system. In the upper tracheae of infected hamsters, although epithelial cells were sporadically positive for viral N protein at 1 d.p.i. irrespective of the inoculum, the N-protein positivity became undetectable at 3 d.p.i. (Fig. 4a). In addition, the viral RNA loads in the upper tracheae of all of the infected hamsters that were tested decreased over time (Extended Data Fig. 5a), suggesting that all of the SARS-CoV-2 isolates used in this study—including Omicron—grow less efficiently in the upper tracheal tissues of hamsters. On the other hand, in lung specimens at 1 d.p.i., B.1.1 virus and Delta infections exhibited strong positivity for the SARS-CoV-2 N protein, and this was similar for the bronchial epithelium of the main bronchus in the lung hilum (Fig. 4b). By contrast, in

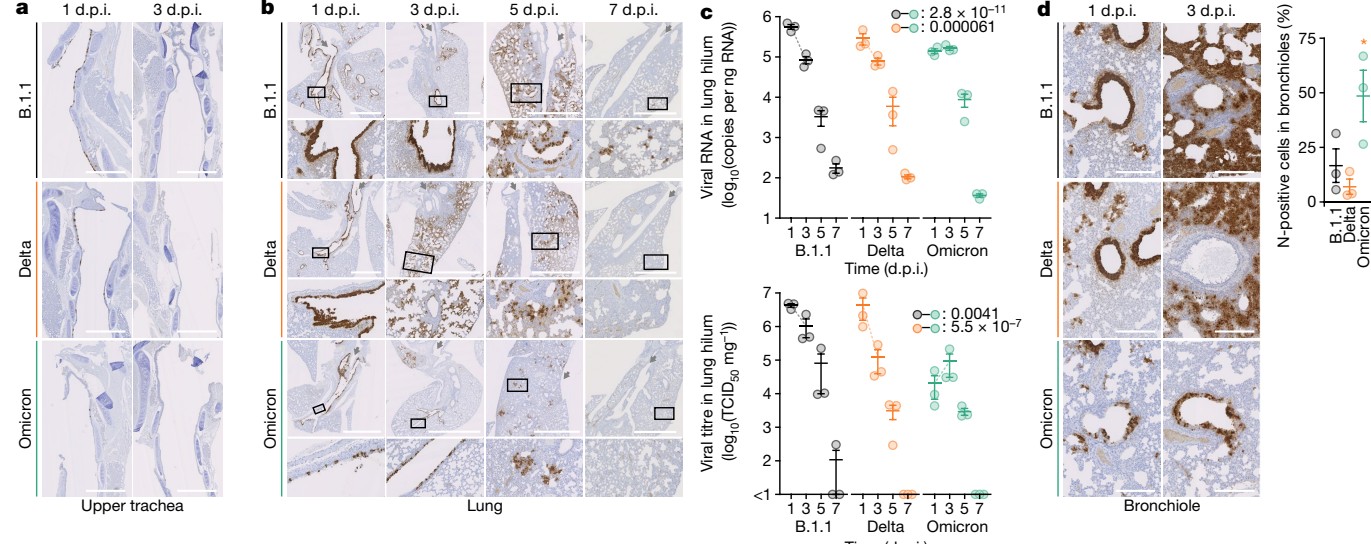

**Fig. 4 | Virological features of Omicron in vivo.** Syrian hamsters were intranasally inoculated with B.1.1 (*n* = 3), Delta (*n* = 3) or Omicron (*n* = 3). **a**, **b**, IHC of the SARS-CoV-2 N protein in the upper trachea and the lungs of infected hamsters. Representative IHC panels of the viral N proteins in the upper part of the trachea from the oral entrance at the vertical levels of thyroid cartilage (**a**) and the lungs (**b**) of infected hamsters. Grey arrows in **b** indicate the bronchus of each lung lobe, and higher-magnification views of the regions indicated by squares are shown at the bottom. Scale bars, 1 mm (**a**); 2.5 mm (**b**). **c**, Quantification of viral RNA load (top) and viral titre (50% tissue culture infectious dose (TCID$_{50}$); bottom) in the lung hilum. Broken lines indicate the slopes between 1 and 3 d.p.i. **d**, IHC of viral N protein in the bronchioles in the vicinity of the lung hilum. Left, representative IHC panels of the viral N proteins. Scale bars, 250 μm. Right, percentage of N-positive cells in bronchiole at 3 d.p.i. Values were measured as described in the Methods. Raw data are shown in Extended Data Fig. 7. In **c**, **d**, data are mean ± s.e.m., and each dot indicates the result from an individual hamster. Statistically significant differences of the slopes were determined by a likelihood-ratio test comparing the models with or without the interaction term of time point and inoculum. FWERs calculated using the Holm method are indicated. In **d**, statistically significant differences (**P* < 0.05) versus B.1.1 and Delta were determined by two-sided unpaired Student's *t*-tests without adjustment for multiple comparisons.

Omicron-infected hamsters at 1 d.p.i., N-positive cells were sporadically detected at the lober portion of the main bronchus, and each N-positive cell exhibited only sparse N staining (Fig. 4b). At 3 d.p.i., the N protein was observed in the alveolar space around the bronchi and bronchioles in the B.1.1-infected and Delta-infected hamsters, and the Delta N disappeared from the bronchial epithelium (Fig. 4b). In Omicron-infected hamsters, the positivity for N protein was not observed in the main bronchial epithelium but remained in the periphery of the bronchi and bronchioles (Fig. 4b). At 5 d.p.i., B.1.1 and Delta N-positive cells were prominently distributed in the alveolar space, whereas only sparse and weakly stained N-positive cell clusters were detected in lungs infected with Omicron (Fig. 4b). At 7 d.p.i., N-positive cells remained sporadically in the alveoli of B.1.1-infected hamsters, whereas few and faintly stained cells were found in the Delta- and Omicron-infected specimens (Fig. 4b). These data suggest that although the B.1.1 virus and Delta efficiently infect the bronchial epithelium and invade the alveolar space, Omicron infects only a portion of the bronchial epithelial cells and is less efficiently transmitted to the neighbouring epithelial cells. Overall, the IHC data suggest that Omicron infection spreads relatively slowly from the main bronchus to the distal portion of the bronchioles, which results in the sporadic distribution of weakly N-positive clusters in the lung alveolar space of hamsters infected with Omicron.

Next, the lungs were resected and separated into two regions—the hilum and the periphery—at different time points (Extended Data Fig. 6). In the lung periphery, the dynamics of viral spread of B.1.1, Delta and Omicron at 1–3 d.p.i. showed similar patterns (Extended Data Fig. 5b). On the other hand, in the lung hilum, although the values of viral RNA load and viral titre of B.1.1 and Delta at 3 d.p.i. were approximately 10-fold lower than those at 1 d.p.i., for Omicron these values at 3 d.p.i. were comparable to—or even higher than—those at 1 d.p.i. (Fig. 4c). Our statistical analysis showed that the slopes of viral RNA and viral titre from 1 to 3 d.p.i. for Omicron were significantly different from those of B.1.1 and Delta (Fig. 4c). These results raise the possibility that the growth dynamics of Omicron during the acute phase of infection, particularly at 1–3 d.p.i., are different from those of B.1.1 and Delta in the lung hilum. To address this possibility in depth, we investigated the positivity for N protein, particularly focusing on the bronchioles that are included in the lung area close to the hilum. The bronchiolar epithelial cells were relatively strongly positive for viral N antigen at 1 d.p.i. (Fig. 4d). At 3 d.p.i., the number of N-positive epithelial cells decreased in B.1.1-infected hamsters compared with that at 1 d.p.i., and most of the bronchiolar epithelial cells became negative for the N protein in Delta-infected hamsters (Fig. 4d). Conversely, the N-positive epithelial cells remained in Omicron-infected hamsters at 3 d.p.i. (Fig. 4d). Furthermore, a quantitative analysis showed that, at 3 d.p.i., the percentage of N-positive cells in the bronchioles of Omicron-infected hamsters was significantly higher than that in Delta-infected hamsters (Fig. 4d, Extended Data Fig. 7). Overall, these results—that is, the positivity for viral N protein in the bronchioles in the vicinity of the lung hilum (Fig. 4d)—correspond well with the viral RNA load and viral titre in the lung hilum (Fig. 4c), as well as the viral RNA load in oral swabs (Fig. 3e).

## Pathological features of Omicron

To further investigate the pathogenicity of Omicron in the lung, the formalin-fixed right lungs of infected hamsters were analysed by carefully identifying the four lobes and main bronchus and lobar bronchi, and sectioning each lobe along with the bronchial branches (Extended Data Fig. 6). In B.1.1-infected and Delta-infected lungs, inflammatory reactions peaked at 5 d.p.i., and inflammation with type II alveolar pneumocyte hyperplasia was found to be widely distributed throughout each lobe (Fig. 5a, Extended Data Fig. 8). By contrast, Omicron infection was associated with limited inflammatory nodules along with the

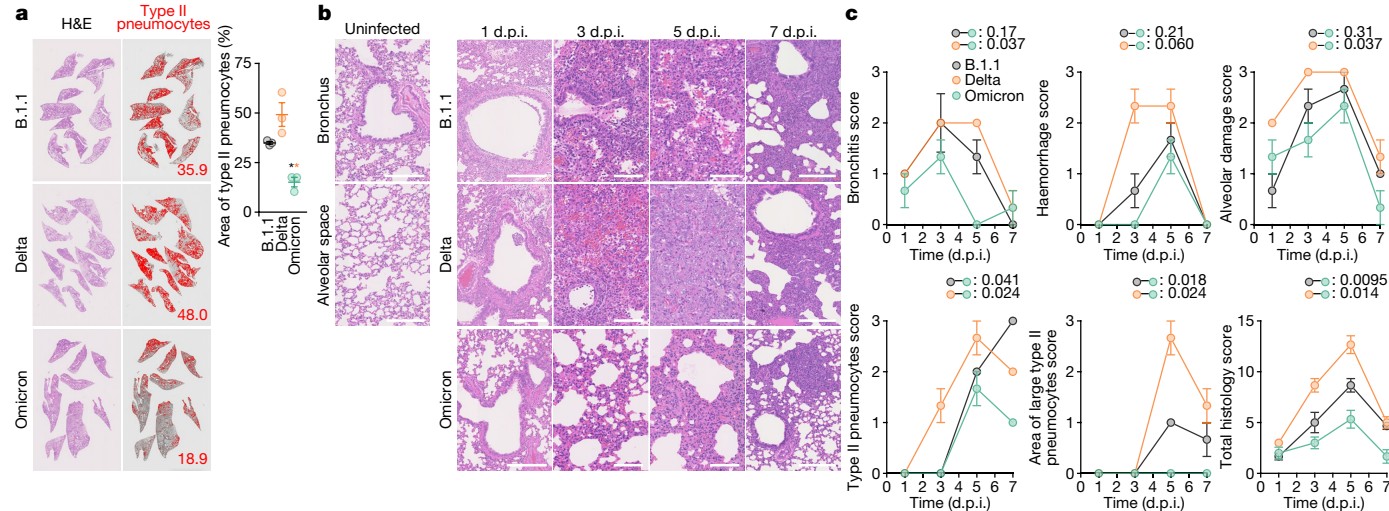

**Fig. 5 | Pathological features of Omicron.** Syrian hamsters were intranasally inoculated with B.1.1 (*n* = 3), Delta (*n* = 3) or Omicron (*n* = 3). **a**, **b**, Histopathological features of lung lesions. Lung sections from infected hamsters were stained with haematoxylin and eosin (H&E). **a**, Section of all four lung lobes at 5 d.p.i. In the middle panels, the inflammatory area with type II pneumocytes is indicated in red. The number in the panel indicates the percentage of the section represented by the indicated area. Right, summary of the percentage of the section represented by type II pneumocytes (3 hamsters per group). Raw data are shown in Extended Data Fig. 8. **b**, H&E staining of the lungs of infected hamsters. Uninfected lung alveolar space and bronchioles at the same time point (Fig. 5a, Extended Data Fig. 8), and the percentage of the area of type II pneumocyte hyperplasia in the Omicron-infected lungs was significantly lower than that in the other two infection groups (Fig. 5a). In the B.1.1-infected hamsters, mild bronchitis was found at 1 d.p.i.; disruptions of bronchi and bronchioles were observed at 3 d.p.i.; and alveolitis and haemorrhage were recognized at 5 d.p.i. at the peak of inflammation (Fig. 5b, c). In the Delta-infected hamsters, the inflammatory reaction was more prominent than in the B.1.1 virus infection and, as shown previously[2], hyperplastic large type II pneumocytes were observed at 5 d.p.i.; at 7 d.p.i., acute inflammatory features (such as bronchitis or bronchiolitis and haemorrhage) were resolved and replaced by type II pneumocytes in these two infection groups (Fig. 5b, c). The observations in these two infection groups correspond well with our previous report[2]. In the Omicron-infected hamsters, mild bronchitis was observed at 1 d.p.i., and at 3 d.p.i., a vague thickening of the alveolar septa and the peribronchial or peribronchiolar nodular distribution of type II pneumocytes were observed (Fig. 5b, c). Notably, severe alveolitis and haemorrhage were not observed in the lungs of Omicron-infected hamsters. At 7 d.p.i., the area of nodular type II pneumocytes was decreased (Fig. 5b, c). Lung lesions were also quantitatively evaluated by histopathological scoring. The total score of Omicron-infected hamsters was significantly lower than that of the B.1.1-infected and Delta-infected hamsters, and each index—such as bronchitis, alveolitis, type II pneumocyte hyperplasia and large type II pneumocyte hyperplasia—was significantly lower in Omicron-infected hamsters than in Delta-infected hamsters (Fig. 5c). Together with the time-course observations (Fig. 3a–d), our results suggest that Omicron is relatively less pathogenic than Delta and the B.1.1 virus.

bronchioles are shown (left). Scale bars, 250 μm (uninfected lung alveolar space and bronchioles and infected hamsters at 1 and 7 d.p.i.); 100 μm (infected hamsters at 3 and 5 d.p.i.). **c**, Histopathological scoring of lung lesions. Representative pathological features are shown in our previous study[2]. Data are mean ± s.e.m. (**a**, **c**). In **a**, each dot indicates the result from an individual hamster. Statistically significant differences (**P* < 0.05) versus B.1.1 and Delta were determined by two-sided unpaired Student's *t*-tests without adjustment for multiple comparisons. In **c**, statistically significant differences versus B.1.1 and Delta through time points were determined by multiple regression. FWERs calculated using the Holm method are indicated.

## Discussion

Recent studies, including ours, have revealed the pronounced resistance of the SARS-CoV-2 Omicron variant against immunity elicited by previous infections and vaccination[24–29]. Here we show that Omicron is less pathogenic than Delta and its ancestral early-pandemic variant (B.1.1 lineage)[14] in a hamster model. Although it is not certain that the viral dynamics in infected hamsters will completely mirror those in humans, our results in an experimental hamster model suggest that the decreased viral spread in the lung tissues is one of the reasons for the attenuated pathogenicity of Omicron. Because Omicron (B.1.1.529 and BA lineages) is phylogenetically classified as a descendant in the B.1.1 lineage[14], our data suggest that Omicron has evolved decreased pathogenicity.

We show that Omicron is less replicative than an early-pandemic SARS-CoV-2 variant and the Delta variant in cell cultures. This might appear contradictory to the rapid rate of spread of Omicron in human society. However, consistent with our previous report[2], the growth of Delta—which surpassed other variants and was the dominant causative agent of the SARS-CoV-2 pandemic in January 2022—was not higher than that of an early-pandemic strain of SARS-CoV-2, suggesting that the growth capacity of SARS-CoV-2 in cell cultures does not necessarily reflect rapid viral spread in society. Rather, we showed here that the dynamics of viral RNA load in oral swabs of Omicron-infected hamsters during the acute phase of infection are different from those of B.1.1- and Delta-infected hamsters. These dynamics correspond to those of viral RNA load and viral titre in the lung hilum as well as positivity for viral N protein in the bronchiolar epithelial cells in the vicinity of the lung hilum of Omicron-infected hamsters. These data suggest that Omicron-infected cells that are retained in the bronchiolar epithelia in the vicinity of the lung hilum could be a major source for the viruses excreted to the oral cavity at 3 d.p.i. The differences in the dynamics of viral excretion to the oral cavity and the infection tropism of Omicron compared with B.1.1 and Delta may perhaps partially explain the rapid spread of Omicron in the human population.

Although the crystal structure of Omicron S is has been determined[30], the molecular and structural mechanisms that underlie how Omicron S is resistant to furin-mediated cleavage remain unclear. However, when we compared the three SARS-CoV-2 isolates used in this study—Omicron, Delta and an early-pandemic SARS-CoV-2 (the B.1.1 virus)—the efficacy of S cleavage, fusogenicity and pathogenicity were associated with each other. The association between S

cleavage efficacy and viral pathogenicity is reminiscent of findings in furin cleavage site (FCS)-deficient SARS-CoV-2; a previous study showed that the FCS-deleted virus exhibits reduced S protein processing in cell cultures and attenuated pathogenicity in experimental animal models[31]. Although the fusogenicity of the FCS-deleted virus has not yet been evaluated, the association between higher viral fusogenicity and greater viral pathogenicity has been reported in other viral infections such as HIV-1[32] and measles[33,34]. Furthermore, whereas the greater severity of COVID-19 and unusual symptoms that are caused by Delta infection have been well documented[35–37], a reduced risk of severe COVID-19 in individuals who are infected with Omicron compared to those infected with Delta has been recently reported[38]. Therefore, the fusogenicity and S1/S2 cleavage efficacy of SARS-CoV-2 may be linked to the degree of its pathogenicity.

The attenuated pathogenicity of Omicron might be considered good news for human society, because such emerging variants pose less of a threat in terms of disease progression. However, as shown in this study and others[22], Omicron spreads more rapidly than Delta; and moreover, Omicron appears to be much more resistant to vaccine-induced immunity than other SARS-CoV-2 variants, including Delta[24–29]. We should note that viral pathogenicity has a linear effect on the increase in hospital admissions, severe cases and deaths, whereas the rate at which the virus spreads in the human population has an exponential effect on these factors. Therefore, we cannot conclude that the risk of Omicron for global health is relatively low, and we suggest that this SARS-CoV-2 variant should continue to be monitored in depth.

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

**The Genotype to Phenotype Japan (G2P-Japan) Consortium**

**Mai Suganami[2], Akiko Oide[2], Mika Chiba[2], Hayato Ito[1], Tomokazu Tamura[1], Kana Tsushima[1], Haruko Kubo[1], Zannatul Ferdous[4], Hiromi Mouri[4], Miki Iida[4], Keiko Kasahara[4], Koshiro Tabata[6], Mariko Ishizuka[6], Asako Shigeno[18], Kenzo Tokunaga[20], Seiya Ozono[20], Isao Yoshida[17], So Nakagawa[21], Jiaqi Wu[21], Miyoko Takahashi[21], Atsushi Kaneda[22], Motoaki Seki[22], Ryoji Fujiki[22], Bahityar Rahmutulla Nawai[22], Yutaka Suzuki[23], Yukie Kashima[23], Kazumi Abe[23], Kiyomi Imamura[23], Kotaro Shirakawa[24], Akifumi Takaori-Kondo[24], Yasuhiro Kazuma[24], Ryosuke Nomura[24], Yoshihito Horisawa[24], Kayoko Nagata[24], Yugo Kawai[24], Yohei Yanagida[24], Yusuke Tashiro[24], Otowa Takahashi[9], Kazuko Kitazato[9], Haruyo Hasebe[9], Chihiro Motozono[25], Mako Toyoda[25], Toong Seng Tan[25], Isaac Ngare[25], Takamasa Ueno[25], Akatsuki Saito[26], Erika P. Butlertanaka[26], Yuri L. Tanaka[26] & Nanami Morizako[26]**

[20]National Institute of Infectious Diseases, Tokyo, Japan. [21]Tokai University, Isehara, Japan. [22]Chiba University, Chiba, Japan. [23]The University of Tokyo, Kashiwa, Japan. [24]Kyoto University, Kyoto, Japan. [25]Kumamoto University, Kumamoto, Japan. [26]University of Miyazaki, Miyazaki, Japan.

## Methods

### Ethics statement

All experiments with hamsters were performed in accordance with the Science Council of Japan's Guidelines for the Proper Conduct of Animal Experiments. The protocols were approved by the Institutional Animal Care and Use Committee of National University Corporation Hokkaido University (approval numbers 20-0123 and 20-0060).

### Omicron epidemiological and viral sequence data

The seven-day average of new COVID-19 cases per day in South Africa and the UK through 24 December 2021 were downloaded from Our World in Data (https://ourworldindata.org/covid-cases) on 4 January 2022. The numbers of Omicron sequences reported and the countries that had reported Omicron sequences as of 7 January 2022 were obtained from outbreak.info (https://outbreak.info) on 10 January 2022.

### Modelling the dynamics of SARS-CoV-2 lineages

To compare the viral spread rate in the human population of each SARS-CoV-2 lineage, we estimated the relative effective reproduction number of each viral lineage according to the lineage dynamics calculated on the basis of viral genomic surveillance data. The data were downloaded from the GISAID database (https://www.gisaid.org/) on January 4, 2022. We analysed the datasets of the seven countries with more than 1,500 Omicron sequences (South Africa, Australia, Denmark, Germany, Israel, the UK and the USA) (Fig. 1, Extended Data Fig. 1). The dynamics of the five most predominant lineages in each country from 1 January 2021 to 24 December 2021, were analysed except for the USA. In the case of the USA, the six most predominant lineages in that period were analysed because Omicron was the sixth predominant lineage in this country.

We prepared the input data to estimate the relative effective reproduction number of each viral lineage for each country on the basis of the metadata of the sequenced SARS-CoV-2 strains (that is, the collection date, collection place and PANGO lineage) provided from the GISAID database (https://www.gisaid.org/). The viral strains belonging to the predominant lineages were used for the subsequent analysis. The number of strains in each viral lineage isolated on each day was counted and subsequently summed in three-day bins. Finally, the count matrix representing the abundance of the respective viral lineages [viral lineage ID $k \in \{1, 2, ..., K\}$; $K = 5$ (for South Africa, Australia, Denmark, Germany, Israel and the UK) or 6 (for the USA)] in the respective time bins ($t \in \{1, 2, ..., T\}$; $T = 119$) for each country was constructed.

We constructed a Bayesian statistical model to represent the transition of the relative frequency of $K$ types of viral lineages with a Bayesian multinomial logistic regression, which is conceptually similar to the models used in previous studies[19-21]. The model is:

$$\boldsymbol{\mu}_t = \mathbf{b}_0 + \mathbf{b}_1 t$$

$$\boldsymbol{\theta}_t = \mathrm{softmax}(\boldsymbol{\mu}_t)$$

$$N_t = \sum_{1 \le k \le K} \mathbf{Y}_{tk}$$

$$\mathbf{Y}_t \sim \mathrm{Multinomial}(N_t, \boldsymbol{\theta}_t)$$

in which $\mathbf{b}_0$, $\mathbf{b}_1$, $\boldsymbol{\mu}_t$, $\boldsymbol{\theta}_t$ and $\mathbf{Y}_t$ are vectors with $K$ elements, and the $k$-th element in the vector represents the value for viral lineage $k$. The explanatory variable is time bin $t$, and the outcome variable $\mathbf{Y}_t$ represents the counts of the respective viral lineages at time $t$. In the model, the linear estimator $\boldsymbol{\mu}_t$, consisting of the intercept $\mathbf{b}_0$ and the slope $\mathbf{b}_1$ for $t$, is converted to the simplex $\boldsymbol{\theta}_t$, which represents the probability of occurrence of each viral lineage, by the softmax link function defined as:

$$\mathrm{softmax}(\mathbf{x}) = \frac{\exp(\mathbf{x})}{\sum_{1 \le j \le J} \exp(\mathbf{x}_j)}.$$

$\mathbf{Y}_t$ is generated from $\boldsymbol{\theta}_t$, and $N_t$, which represents the total count of all lineages at $t$, according to a multinomial distribution.

The relative effective reproduction number of each viral lineage ($\mathbf{r}$, a vector with $K$ elements) was calculated according to the slope parameter $\mathbf{b}_1$ in the model above with the assumption of a fixed generation time. According to the previous study[19], the relative effective reproduction number $\mathbf{r}$ was defined as:

$$\mathbf{r} = \exp(\gamma/w\mathbf{b}_1),$$

in which $\gamma$ is the average viral generation time (5.5 days)[39] and $w$ is the time bin size (3 days). For the parameter estimation, the intercept and slope parameters of the Delta variant were fixed at 0. Consequently, the relative effective reproduction number of Delta was fixed at 1, and those of the respective lineages were estimated relative to that of Delta.

Parameter estimation was performed by the framework of Bayesian statistical inference with Markov chain Monte Carlo (MCMC) methods implemented in CmdStan v.2.28.1 (https://mc-stan.org) with cmdstanr v.0.4.0 (https://mc-stan.org/cmdstanr/). Noninformative priors were set for all parameters. Four independent MCMC chains were run with 2,000 and 4,000 steps in the warmup and sampling iterations, respectively. In the MCMC runs, the target average acceptance probability was set at 0.99, and the maximum tree depth exceeded was set at 20. We confirmed that all estimated parameters had <1.01 $\hat{R}$ convergence diagnostic and more than 1,000 effective sampling size values, indicating that the MCMC runs were successfully convergent. The fitted model closely recapitulated the observed viral lineage dynamics in each country ($R^2 > 0.99$ in all countries; Extended Data Fig. 1c) The analyses above were performed in R v.3.6.3 (https://www.r-project.org/).

### Cell culture

HEK293 cells (a human embryonic kidney cell line; ATCC CRL-1573) and HEK293-ACE2/TMPRSS2 cells (HEK293 cells (ATCC CRL-1573) stably expressing human ACE2 and TMPRSS2)[23] were maintained in Dulbecco's modified Eagle's medium (DMEM) (high-glucose) (Wako, 044-29765) containing 10% fetal bovine serum (FBS) and 1% penicillin–streptomycin (PS). A549 (a human lung epithelial cell line; ATCC CCL-185) and A549-ACE2 cells (A549 cells (ATCC CCL-185) stably expressing human ACE2)[23] were maintained in Ham's F-12K medium (Wako, 080-08565) containing 10% FBS and 1% PS. Vero cells (an African green monkey (*Chlorocebus sabaeus*) kidney cell line; JCRB0111) were maintained in Eagle's minimum essential medium (EMEM) (Wako, 051-07615) containing 10% FBS and 1% PS. VeroE6/TMPRSS2 cells (VeroE6 cells stably expressing human TMPRSS2; JCRB1819)[40] were maintained in DMEM (low-glucose) (Wako, 041-29775) containing 10% FBS, G418 (1 mg ml$^{-1}$; Nacalai Tesque, G8168-10ML) and 1% PS. Calu-3 cells (a human lung epithelial cell line; ATCC HTB-55) were maintained in EMEM (Sigma-Aldrich, M4655-500ML) containing 20% FBS and 1% PS. Calu-3/DSP$_{1-7}$ cells (Calu-3 cells (ATCC HTB-55) stably expressing DSP$_{1-7}$)[41] were maintained in EMEM (Wako, 056-08385) supplemented with 20% FBS and 1% PS. HeLa-ACE2/TMPRSS2 cells (HeLa229 cells (JCRB9086) stably expressing human ACE2 and TMPRSS2)[42] were maintained in DMEM (low-glucose) (Wako, 041-29775) containing 10% FBS, G418 (1 mg ml$^{-1}$; Nacalai Tesque, G8168-10ML) and 1% PS. All cell lines were regularly tested for mycoplasma contamination by PCR and were confirmed to be mycoplasma-free. Primary human nasal epithelial cells (EP02, batch MP0010) were purchased from Epithelix and maintained according to the manufacturer's instructions.

## SARS-CoV-2 preparation and titration

An Omicron variant (BA.1 lineage, strain TY38-873; GISAID ID: EPI_ISL_7418017)[43] was obtained from the National Institute of Infectious Diseases, Japan. An early-pandemic D614G-bearing isolate (B.1.1 lineage, strain TKYE610670; GISAID ID: EPI_ISL_479681) and a Delta isolate (B.1.617.2 lineage, strain TKYTK1734; GISAID ID: EPI_ISL_2378732) were used in the previous study[2].

Virus preparation and titration was performed as previously described[2,23]. To prepare the working virus stock, 20 μl of the seed virus was inoculated into VeroE6/TMPRSS2 cells ($5 \times 10^6$ cells in a T-75 flask). One hour after infection, the culture medium was replaced with DMEM (low-glucose) (Wako, 041-29775) containing 2% FBS and 1% PS. At 3 d.p.i., the culture medium was collected and centrifuged, and the supernatants were collected as the working virus stock. The viral genome sequences of working viruses were verified as described below.

The titre of the prepared working virus was measured as the 50% tissue culture infectious dose ($TCID_{50}$). In brief, one day before infection, VeroE6/TMPRSS2 cells (10,000 cells) were seeded into a 96-well plate. Serially diluted virus stocks were inoculated into the cells and incubated at 37 °C for four days. The cells were observed under microscopy to judge the cytopathic effect appearance. The value of $TCID_{50}$ $ml^{-1}$ was calculated with the Reed–Muench method[44].

## SARS-CoV-2 infection

One day before infection, Vero cells (10,000 cells), VeroE6/TMPRSS2 cells (10,000 cells), Calu-3 cells (20,000 cells), HeLa-ACE2/TMPRSS2 cells (10,000 cells), A549-ACE2 cells (10,000 cells) and A549 cells (10,000 cells) were seeded into a 96-well plate. SARS-CoV-2 (100 $TCID_{50}$ for VeroE6/TMPRSS2 cells (Extended Data Fig. 2); 1,000 $TCID_{50}$ for Vero cells (Fig. 2a), VeroE6/TMPRSS2 cells (Fig. 2a), A549-ACE2 cells (Fig. 2a), HeLa-ACE2/TMPRSS2 cells (Extended Data Fig. 2) and A549 cells (Extended Data Fig. 2); and 2,000 $TCID_{50}$ for Calu-3 cells (Fig. 2a)) was inoculated and incubated at 37 °C for 1 h. The infected cells were washed, and 180 μl of culture medium was added. The culture supernatant (10 μl) was collected at the indicated time points and used for RT–qPCR to quantify the viral RNA copy number (see below). To monitor the syncytium formation in infected cell culture, bright-field photos were obtained using an All-in-One Fluorescence Microscope BZ-X800 (Keyence).

The infection experiment in primary human nasal epithelial cells (Fig. 2a) was performed as previously described[2]. In brief, the working viruses were diluted with Opti-MEM (Thermo Fisher Scientific, 11058021). The diluted viruses (1,000 $TCID_{50}$ in 100 μl) were inoculated onto the apical side of the culture and incubated at 37 °C for 1 h. The inoculated viruses were removed and washed twice with Opti-MEM. To collect the viruses on the apical side of the culture, 100 μl Opti-MEM was applied onto the apical side of the culture and incubated at 37 °C for 10 min. The Opti-MEM applied was collected and used for RT–qPCR to quantify the viral RNA copy number (see below).

## Immunofluorescence staining

Immunofluorescence staining was performed as previously described[2]. In brief, one day before infection, VeroE6/TMPRSS2 cells (10,000 cells) were seeded into 96-well, glass bottom, black plates and infected with SARS-CoV-2 (100 $TCID_{50}$). At 24 h.p.i., the cells were fixed with 4% paraformaldehyde in phosphate-buffered saline (PBS) (Nacalai Tesque, 09154-85) for 1 h at 4 °C. The fixed cells were permeabilized with 0.2% Triton X-100 in PBS for 1 h and blocked with 10% FBS in PBS for 1 h at 4 °C. The fixed cells were then stained using rabbit anti-SARS-CoV-2 N polyclonal antibody (GeneTex, GTX135570, 1:1,000) for 1 h. After washing three times with PBS, cells were incubated with an Alexa 488-conjugated anti-rabbit IgG antibody (Thermo Fisher Scientific, A-11008, 1:1,000) for 1 h. Nuclei were stained with DAPI (Thermo Fisher Scientific, 62248). Fluorescence microscopy was performed on an All-in-One Fluorescence Microscope BZ-X800 (Keyence).

## Plaque assay

The plaque assay was performed as previously described[2,23]. In brief, one day before infection, VeroE6/TMPRSS2 cells (100,000 cells) were seeded into a 24-well plate and infected with SARS-CoV-2 (10,000 $TCID_{50}$) at 37 °C. At 2 h.p.i., mounting solution containing 3% FBS and 1.5% carboxymethyl cellulose (Wako, 039-01335) was overlaid, followed by incubation at 37 °C. At 3 d.p.i., the culture medium was removed, and the cells were washed with PBS three times and fixed with 4% paraformaldehyde phosphate (Nacalai Tesque, 09154-85). The fixed cells were washed with tap water, dried and stained with staining solution (0.1% methylene blue (Nacalai Tesque, 22412-14) in water) for 30 min. The stained cells were washed with tap water and dried, and the size of plaques was measured using Fiji software v.2.2.0 (ImageJ).

## RT–qPCR

RT–qPCR was performed as previously described[2,23]. In brief, 5 μl of culture supernatant was mixed with 5 μl of 2× RNA lysis buffer (2% Triton X-100, 50 mM KCl, 100 mM Tris-HCl (pH 7.4), 40% glycerol and 0.8 U μl$^{-1}$ recombinant RNase inhibitor (Takara, 2313B)) and incubated at room temperature for 10 min. RNase-free water (90 μl) was added, and the diluted sample (2.5 μl) was used as the template for real-time RT–PCR performed according to the manufacturer's protocol using the One Step TB Green PrimeScript PLUS RT-PCR kit (Takara, RR096A) and the following primers: forward N, 5'-AGCCTCTTCTCGTTCCTCATCAC-3'; and reverse N, 5'-CCGCCATTGCCAGCCATTC-3'. The viral RNA copy number was standardized with a SARS-CoV-2 direct detection RT–qPCR kit (Takara, RC300A). Fluorescent signals were acquired using a Quant-Studio 3 Real-Time PCR system (Thermo Fisher Scientific), CFX Connect Real-Time PCR Detection system (Bio-Rad), Eco Real-Time PCR System (Illumina), qTOWER3 G Real-Time System (Analytik Jena) or 7500 Real-Time PCR System (Thermo Fisher Scientific).

## Plasmid construction

Plasmids expressing the SARS-CoV-2 S proteins of the D614G-bearing early-pandemic SARS-CoV-2 (pC-SARS2-S D614G) and Delta (pC-SARS2-S Delta) were prepared in our previous study[2,23]. A plasmid expressing the SARS-CoV-2 Omicron S protein (pC-SARS2-S Omicron) was generated by overlap extension PCR using pC-SARS2-S D614G[2,23] and pC-SARS2-S Alpha[2] as the templates and the primers listed in Supplementary Table 2. The resulting PCR fragment was digested with KpnI and NotI and inserted into the KpnI-NotI site of the pCAGGS vector. The sequence of constructed plasmid was verified by using Sequencher software v.5.1 (Gene Codes Corporation).

## SARS-CoV-2 S-based fusion assay

The SARS-CoV-2 S-based fusion assay was performed as previously described[2,23]. This assay uses a dual split protein (DSP) encoding *Renilla* luciferase and *GFP* genes; the respective split proteins, $DSP_{8-11}$ and $DSP_{1-7}$, are expressed in effector and target cells by transfection. In brief, on day 1, effector cells (that is, S-expressing cells) and target cells (see below) were prepared at a density of $0.6–0.8 \times 10^6$ cells in a 6-well plate. To prepare effector cells, HEK293 cells were cotransfected with the S expression plasmids (400 ng) and $pDSP_{8-11}$ (400 ng) using TransIT-LT1 (Takara, MIR2300). To prepare target cells, HEK293 cells were cotransfected with pC-ACE2 (200 ng) and $pDSP_{1-7}$ (400 ng). Target HEK293 cells in selected wells were cotransfected with pC-TMPRSS2 (40 ng) in addition to the plasmids above. VeroE6/TMPRSS2 cells were transfected with $pDSP_{1-7}$ (400 ng). On day 3 (24 h post-transfection), 16,000 effector cells were detached and reseeded into 96-well black plates (PerkinElmer, 6005225), and target cells (HEK293, VeroE6/TMPRSS2 or Calu-3/$DSP_{1-7}$ cells) were reseeded at a density of 1,000,000 cells per 2 ml per well in 6-well plates. On day 4 (48 h post-transfection), target cells were incubated with EnduRen live cell substrate (Promega, E6481) for 3 h and then detached, and 32,000 target cells were

added to a 96-well plate with effector cells. *Renilla* luciferase activity was measured at the indicated time points using Centro XS3 LB960 (Berthhold Technologies). To measure the surface expression level of S protein, effector cells were stained with rabbit anti-SARS-CoV-2 S S1/S2 polyclonal antibody (Thermo Fisher Scientific, PA5-112048, 1:100) or mouse anti-SARS-CoV-2 S monoclonal antibody (clone 1A9, GeneTex, GTX632604, 1:100). Normal rabbit IgG (SouthernBiotech, 0111-01, 1:100) or purified mouse IgG1 isotype control antibody (clone MG1-45, BioLegend, 401401, 1:100) was used as a negative control, and APC-conjugated goat anti-mouse or anti-rabbit IgG polyclonal antibody (Jackson ImmunoResearch, 115-136-146, 1:50 or 111-136-144, 1:50) was used as a secondary antibody. The surface expression level of S proteins was measured using FACS Canto II (BD Biosciences) and the data were analysed using FlowJo software v,10.7.1 (BD Biosciences). The gating strategy for flow cytometry is shown in Supplementary Fig. 1. To calculate fusion activity, *Renilla* luciferase activity was normalized to the MFI of surface S proteins. The normalized value (that is, *Renilla* luciferase activity per the surface S MFI) is shown as fusion activity.

## Coculture experiment

One day before transfection, effector cells (that is, S-expressing cells) were seeded on the cover glass and put in a 12-well plate, and target HEK293-ACE2/TMPRSS2 cells were prepared at a density of $1.0 \times 10^5$ cells in a 12-well plate. To prepare effector cells, HEK293 cells were cotransfected with the expression plasmids for the parental D614G S, Delta S, Omicron S (500 ng) and pEGFP-C1 (500 ng) using PEI Max (Polysciences, 24765-1). To prepare target cells, HEK293 cells and HEK293-ACE2/TMPRSS2 cells were transfected with pmCherry-C1 (1,000 ng). At 24 h post-transfection, target cells were detached and cocultured with effector cells. At 24 h post-coculture (at 48 h post-transfection), cells were fixed with 4% paraformaldehyde in PBS (Nacalai Tesque, 09154-85) for 15 min at room temperature. Nuclei were stained with Hoechst 33342 (Thermo Fisher Scientific, H3570). The coverslips were mounted on glass slides using Fluoromount-G (Southern Biotechnology, 0100-01) with Hoechst 33342 and observed using an A1Rsi confocal microscope (Nikon). The size of syncytium (yellow area) was measured using Fiji software v.2.2.0 (ImageJ)[45].

## Western blot

Western blotting was performed as previously described[2,23]. For western blots, the HEK293 cells cotransfected with the S expression plasmids and pDSP$_{8-11}$ (see above) (Fig. 2h) and the VeroE6/TMPRSS2 cells infected with SARS-CoV-2 (m.o.i. = 0.01) at 48 h.p.i. (Fig. 2i) were used. To quantify the level of the cleaved S2 protein in the cells, the collected cells were washed and lysed in lysis buffer (25 mM HEPES (pH 7.2), 20% glycerol, 125 mM NaCl, 1% Nonidet P40 substitute (Nacalai Tesque, 18558-54) and protease inhibitor cocktail (Nacalai Tesque, 03969-21)). After quantification of total protein by protein assay dye (Bio-Rad, 5000006), lysates were diluted with 2× sample buffer (100 mM Tris-HCl (pH 6.8), 4% SDS, 12% β-mercaptoethanol, 20% glycerol and 0.05% bromophenol blue) and boiled for 10 min. Then, 10-µl samples (50 µg of total protein) were subjected to western blotting. For protein detection, the following antibodies were used: mouse anti-SARS-CoV-2 S monoclonal antibody (clone 1A9, GeneTex, GTX632604, 1:10,000), rabbit anti-SARS-CoV-2 N monoclonal antibody (clone HL344, GeneTex, GTX635679, 1:5,000), rabbit anti-β-actin (ACTB) monoclonal antibody (clone 13E5, Cell Signalling, 4970, 1:5,000), mouse anti-α-tubulin (TUBA) monoclonal antibody (clone DM1A, Sigma-Aldrich, T9026, 1:10,000), horseradish peroxidase (HRP)-conjugated donkey anti-rabbit IgG polyclonal antibody (Jackson ImmunoResearch, 711-035-152, 1:10,000) and HRP-conjugated donkey anti-mouse IgG polyclonal antibody (Jackson ImmunoResearch, 715-035-150, 1:10,000). Chemiluminescence was detected using SuperSignal West Femto Maximum Sensitivity Substrate (Thermo Fisher Scientific, 34095) or Western BLoT Ultra Sensitive HRP Substrate (Takara, T7104A) according to the manufacturers'

instructions. Bands were visualized using an Amersham Imager 600 (GE Healthcare), and the band intensity was quantified using Image Studio Lite v.5.2 (LI-COR Biosciences) or Fiji software v.2.2.0 (ImageJ). Uncropped blots are shown in Supplementary Fig. 2.

## Animal experiments

Syrian hamsters (male, 4 weeks old) were purchased from Japan SLC and divided into groups by simple randomization. Baseline body weights were measured before infection. For the virus infection experiments, hamsters were anaesthetized by intramuscular injection of a mixture of 0.15 mg kg$^{-1}$ medetomidine hydrochloride (Domitor, Nippon Zenyaku Kogyo), 2.0 mg kg$^{-1}$ midazolam (Dormicum, FUJIFILM Wako Chemicals) and 2.5 mg kg$^{-1}$ butorphanol (Vetorphale, Meiji Seika Pharma). The B.1.1 virus, Delta, Omicron (10,000 TCID$_{50}$ in 100 µl) or saline (100 µl) were intranasally inoculated under anaesthesia. Oral swabs were daily collected under anaesthesia with isoflurane (Sumitomo Dainippon Pharma). Body weight, enhanced pause (Penh, see below), the ratio of time to peak expiratory follow relative to the total expiratory time (Rpef, see below) and subcutaneous oxygen saturation (SpO$_2$, see below) were monitored at 1, 3, 5, 7, 10, and 15 d.p.i. Respiratory organs were anatomically collected at 1, 3, 5 and 7 d.p.i. (for lung) or 1, 3 and 7 d.p.i. (for trachea). Viral RNA load in the oral swabs and respiratory tissues was determined by RT–qPCR. Viral titres in the lung hilum were determined by TCID$_{50}$. These tissues were also used for histopathological and IHC analyses (see below). No method of randomization was used to determine how the animals were allocated to the experimental groups and processed in this study because covariates (sex and age) were identical. The number of investigators was limited, as most of experiments were performed in high-containment laboratories. Therefore, blinding was not carried out.

## Lung function test

Respiratory parameters (Penh and Rpef) were measured by using a whole-body plethysmography system (DSI) according to the manufacturer's instructions. In brief, a hamster was placed in an unrestrained plethysmography chamber and allowed to acclimatize for 30 s, then, data were acquired over a 5-min period by using FinePointe Station and Review software v.2.9.2.12849 (STARR). The state of oxygenation was examined by measuring SpO$_2$ using a pulse oximeter, MouseOx PLUS (STARR). SpO$_2$ was measured by attaching a measuring chip to the neck of hamsters sedated by 0.25 mg kg$^{-1}$ medetomidine hydrochloride.

## H&E staining

H&E staining was performed as described in the previous study[2]. In brief, excised animal tissues were fixed with 10% formalin neutral buffer solution, and processed for paraffin embedding. The paraffin blocks were sectioned with 3-µm thickness and then mounted on silane-coated glass slides (MAS-GP, Matsunami). H&E staining was performed according to a standard protocol.

## IHC

IHC was performed using an Autostainer Link 48 (Dako). The deparaffinized sections were exposed to EnVision FLEX target retrieval solution high pH (Agilent, CK8004) for 20 min at 97 °C to activate, and mouse anti-SARS-CoV-2 N monoclonal antibody (R&D systems, Clone 1035111, MAB10474-SP, 1:400) was used as a primary antibody. The sections were sensitized using EnVision FLEX (Agilent) for 15 min and visualized by peroxidase-based enzymatic reaction with 3,3′-diaminobenzidine tetrahydrochloride as substrate for 5 min.

For the evaluation of the N-protein positivity in the bronchioles in the vicinity of the lung hilum at 3 d.p.i. (Fig. 4d), lung specimens from infected hamsters (B.1.1, Delta and Omicron; $n = 3$ each) were stained with mouse anti-SARS-CoV-2 N monoclonal antibody (R&D systems, clone 1035111, MAB10474-SP, 1:400). All bronchioles were identified by certificated pathologists, and the full length of the circumference of each bronchiole (perimeter) and the length of N-protein positivity

were measured using NDRscan3.2 software (Hamamatsu Photonics). The main lobar bronchus (more than 500 µm in diameter) was excluded from this evaluation. Peripheral branches from lobar bronchus were referred to as bronchioles (less than 500 µm in diameter) and were analysed. The N-protein positivity was calculated as the percentage of the length of N-protein positivity in the full-length bronchioles in each hamster.

### Histopathological scoring of lung lesions

The area of inflammation in the infected lungs (Fig. 5a) was measured by the presence of type II pneumocyte hyperplasia. Three hamsters infected with each virus were euthanized at 5 d.p.i., and all four lung lobes, including right upper (anterior–cranial), middle, lower (posterior–caudal) and accessory lobes, were sectioned along with their bronchi. The tissue sections were stained by H&E, and the digital microscopic images were incorporated into virtual slides using NDRscan3.2 software (Hamamatsu Photonics). The colour of the images was decomposed by RGB in split channels using Fiji software v.2.2.0 (ImageJ).

Histopathological scoring (Fig. 5c) was performed as described in the previous study[2]. In brief, pathological features including bronchitis or bronchiolitis, haemorrhage or congestion, alveolar damage with epithelial apoptosis and macrophage infiltration, hyperplasia of type II pneumocytes, and the area of the hyperplasia of large type II pneumocytes were evaluated by certified pathologists and the degree of these pathological findings were arbitrarily scored using a four-tiered system as 0 (negative), 1 (weak), 2 (moderate) and 3 (severe). The 'large type II pneumocytes' are the hyperplasia of type II pneumocytes exhibiting more than 10-µm-diameter nucleus. We described 'large type II pneumocytes' as one of the notable histopathological features of SARS-CoV-2 infection in our previous study[2]. Total histology score is the sum of these five indices. In the representative lobe of each lung, the inflammation area with type II pneumocytes was gated by the certificated pathologists on H&E staining, and the indicated area was measured by Fiji software v.2.2.0 (ImageJ).

### Viral genome sequencing analysis

The sequences of the working viruses were verified by viral RNA-sequencing analysis. Viral RNA was extracted using the QIAamp viral RNA mini kit (Qiagen, 52906). The sequencing library for total RNA sequencing was prepared using the NEB Next Ultra RNA Library Prep Kit for Illumina (New England Biolabs, E7530). Paired-end, 76-bp sequencing was performed using MiSeq (Illumina) with MiSeq reagent kit v.3 (Illumina, MS-102-3001). Sequencing reads were trimmed using fastp v0.21.0[46] and subsequently mapped to the viral genome sequences of a lineage B isolate (strain Wuhan-Hu-1; GISAID ID: EPI_ISL_402125; GenBank accession no. NC_045512.2) using BWA-MEM v.0.7.17[47]. Variant calling, filtering and annotation were performed using SAMtools v.1.9[48] and snpEff v.5.0e[49].

For the clinical isolates—an Omicron isolate (strain TY38-873; GISAID ID: EPI_ISL_7418017), a Delta isolate (strain TKYTK1734; GISAID ID: EPI_ISL_2378732; ref. [2]) and a D614G-bearing B.1.1 isolate (strain TKYE610670; GISAID ID: EPI_ISL_479681; ref. [2])—the detected variants that are present in the original sequences were excluded. Information on the detected mutations in the working virus stocks is summarized in Supplementary Table 3.

### Statistics and reproducibility

Statistical significance was tested using a two-sided Student's $t$-test or a two-sided Mann–Whitney $U$-test unless otherwise noted. The tests were performed using Excel software v.16.16.8 (Microsoft) or Prism 9 software v.9.1.1 (GraphPad).

In the time-course experiments (Figs. 2a, g, 3a–d, 5c, Extended Data Figs. 2, 3a), a multiple regression analysis including experimental conditions as explanatory variables and time points as qualitative control variables was performed to evaluate the difference between experimental conditions thorough all time points. $P$ value was calculated by a two-sided Wald test. Subsequently, FWERs were calculated by the Holm method. These analyses were performed in R v.3.6.3 (https://www.r-project.org/).

In the time-course data of viral RNA in the oral swab of infected hamsters (Fig. 3e), significant differences in the dynamics between Omicron-infected and B.1.1- or Delta-infected hamsters were determined by a permutation test. In the observed data, the average value at each time point was calculated in each group, and the Euclidean distance of the average dynamics between the two groups was calculated. Next, the permutated data were generated by shuffling the viral group label among hamster individuals for all combinations. As each viral group has six hamsters, a total of $_{12}C_6$ (= 924) combinations of the data were generated. Subsequently, the Euclidean distance of the average dynamics between the two groups was calculated in each permuted data. Finally, the Euclidean distance in each permutated data was compared to that of the observed data, and the $P$ value was calculated by dividing the number of permutated data in which the distance was greater than or equal to that in the observed data by the total number of the permutated data. FWERs were calculated by the Holm method. These analyses were performed in R v3.6.3 (https://www.r-project.org/).

In the hierarchical clustering analysis of infected hamsters based on the dynamics of viral RNA load in the oral swabs (Extended Data Fig. 4), the Euclidean distances of the $\log_{10}$-transformed viral RNA dynamics were calculated among individual hamsters. Subsequently, a dendrogram was reconstructed by Ward's method according to the distance matrix. Clusters were defined by cutting the dendrogram at a height of cluster number = 2. The association between the clustering result and Omicron-infected hamsters was examined by two-sided Fisher's exact test. These analyses were performed in R v.3.6.3 (https://www.r-project.org/).

The slopes of viral RNA load (Fig. 4c, top, Extended Data Fig. 5b) and viral titre (Fig. 4c, bottom) from 1 d.p.i. to 3 d.p.i. were statistically compared between Omicron-infected and B.1.1-infected or Delta-infected hamsters using a likelihood-ratio test. In the likelihood-ratio test, the following full and reduced models were used: the full model included inoculum, time point, and the interaction term of inoculum and time point. The reduced model included inoculum and time point. The $P$ value was calculated by chi-squared test. FWERs were calculated by the Holm method. These analyses were performed in R v.3.6.3 (https://www.r-project.org/).

In Figs. 4a, b, d, 5a, b, Extended Data Figs. 7, 8, the photographs shown are the representative areas of two independent experiments using three hamsters at each time point. In Fig. 2b–d, Extended Data Fig. 3b, assays were performed in triplicate. Photographs shown are the representatives of more than 20 fields of view taken for each sample.

### Reporting summary

Further information on research design is available in the Nature Research Reporting Summary linked to this paper.

## Data availability

The raw data of the viral sequences analysed in this study have been deposited in the Gene Expression Omnibus (accession number: GSE192472). All databases and datasets used in this study are available from GISAID (https://www.gisaid.org), GenBank (https://www.ncbi.nlm.nih.gov/genbank/), Our World in Data (https://ourworldindata.org/covid-cases) or outbreak.info (https://outbreak.info). The accession numbers of the viral sequences used in this study are listed in the Methods section. Source data are provided with this paper.

## Code availability

The computational code to estimate the viral spread rate in the human population (Fig. 1) is available in the GitHub repository (https://github.com/TheSatoLab/Estimation_of_transmissibility_of_each_viral_lineage).

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

**Acknowledgements** We thank all members of The Genotype to Phenotype Japan (G2P-Japan) Consortium. We also thank the National Institute for Infectious Diseases, Japan, for providing an Omicron isolate; A. Sasaki for suggestions; and H. Arase, K. Tokunaga and J. Gohda for providing reagents. The super-computing resource was provided by the Human Genome Center at The University of Tokyo. This study was supported in part by the AMED Research Program on Emerging and Re-emerging Infectious Diseases (20fk0108401 to T.F.; 20fk010847 to T.F.; 21fk0108617 to T.F.; 20fk0108146 to K. Sato; 20fk0108270 to K. Sato; 20fk0108413 to T. Ikeda and K. Sato; and 20fk0108451 to G2P-Japan Consortium, T. Ikeda, T. Irie, K.M., T.F. and K. Sato); AMED Research Program on HIV/AIDS (21fk0410039 to K. Sato); AMED Japan Program for Infectious Diseases Research and Infrastructure (21wm0125008 to H.S. and 21wm0225003 to H.S.); JST A-STEP (JPMJTM20SL to T. Ikeda); JST SICORP (e-ASIA) (JPMJSC20U1 to K. Sato); JST SICORP (JPMJSC21U5 to K. Sato); JST CREST (JPMJCR20H4 to K. Sato); JSPS KAKENHI Grant-in-Aid for Scientific Research B (21H02736 to T.F.; 18H02662 to K. Sato; and 21H02737 to K. Sato); JSPS Fund for the Promotion of Joint International Research (Fostering Joint International Research) (18KK0447 to K. Sato); JSPS Core-to-Core Program (A. Advanced Research Networks) (JPJSCCA20190008, to K. Sato); JSPS Research Fellow DC1 (19J20488, to I.K.); JSPS Leading Initiative for Excellent Young Researchers (LEADER) (to T. Ikeda); World-leading Innovative and Smart Education (WISE) Program 1801 from the Ministry of Education, Culture, Sports, Science and Technology (MEXT) (to N.N.); The Tokyo Biochemical Research Foundation (to K. Sato); Mitsubishi Foundation (to T. Ikeda); Shin-Nihon Foundation of Advanced Medical Research (to T. Ikeda); Tsuchiya Foundation (to T. Irie); an intramural grant from Kumamoto University COVID-19 Research Projects (AMABIE) (to T. Ikeda); Intercontinental Research and Educational Platform Aiming for Eradication of HIV/AIDS (to T. Ikeda); and Joint Usage/Research Center program of Institute for Frontier Life and Medical Sciences, Kyoto University (to K. Sato).

**Author contributions** D.Y., I.K., H.N., K.U., Y.K., R. Shimizu, R.K., T. Ikeda and T. Irie performed cell culture experiments. R. Suzuki, M.K., Y.M., N.N., Y.O., M.S., K. Yoshimatsu, H.S., K.M. and T.F. performed animal experiments. L.W., M.T. and S.T. performed histopathological analysis. H.A., M.N., K. Sadamasu and K. Yoshimura performed viral genome sequencing analysis. J.I. performed statistical, modelling and bioinformatics analyses. T. Ikeda, T. Irie, K.M., S.T., T.F. and K. Sato designed the experiments and interpreted the results. K. Sato wrote the original manuscript. All authors reviewed and proofread the manuscript. The Genotype to Phenotype Japan (G2P-Japan) Consortium contributed to the project administration.

**Competing interests** The authors declare no competing interests.

**Additional information**
**Correspondence and requests for materials** should be addressed to Keita Matsuno, Shinya Tanaka, Takasuke Fukuhara or Kei Sato.

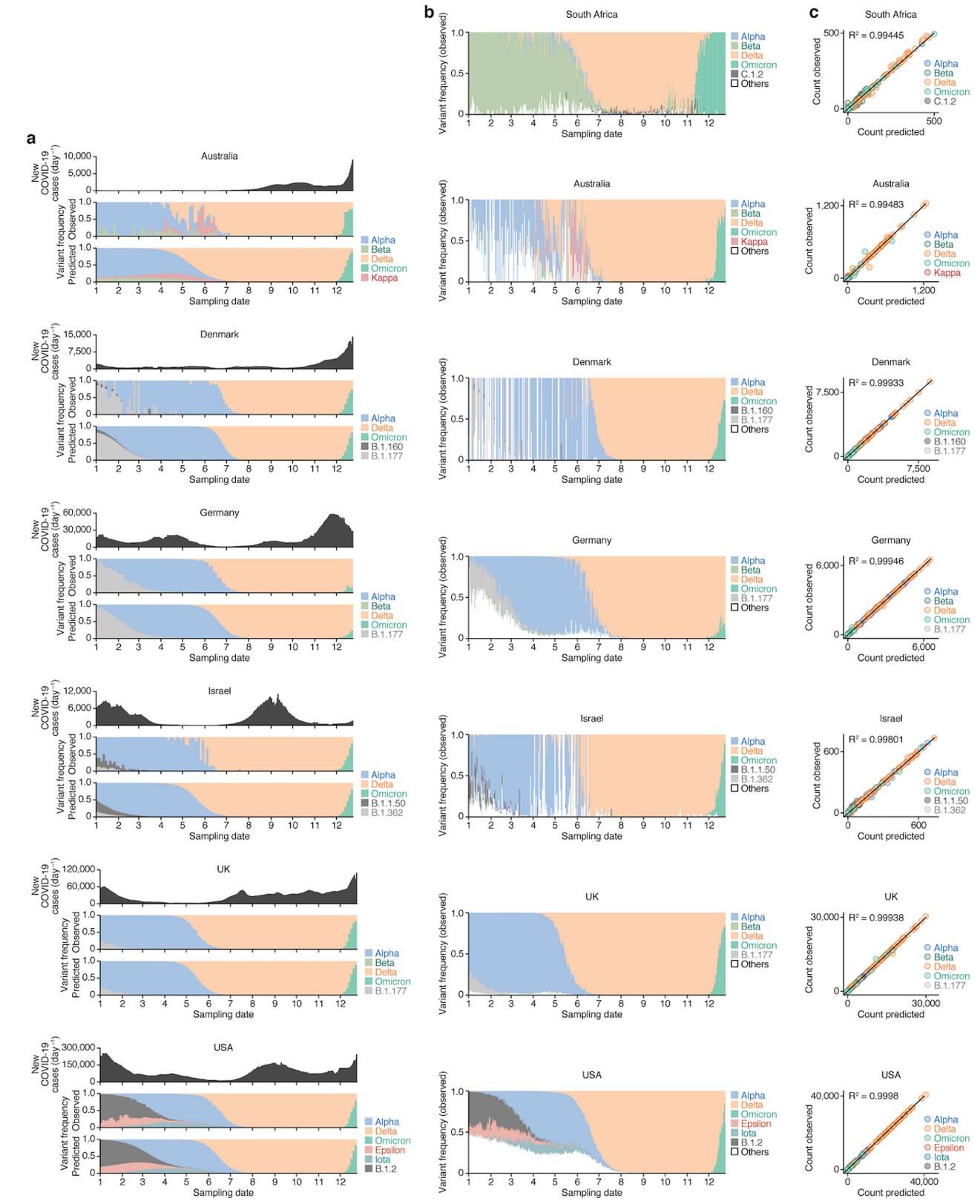

**Extended Data Fig. 1 | Epidemic dynamics of SARS-CoV-2 lineages in seven countries. a**, The 7-day average of new COVID-19 cases reported per day (top), the frequency of the top five (Australia, Denmark, Germany, Israel, and the UK) or six (the USA) viral lineages in the sequenced samples (middle), and the frequency of the viral lineages predicted by our Bayesian statistical model (bottom) are shown. The data in the six countries are from January 1, 2021, to December 24, 2021. The lineage frequency (middle and bottom) is summarized in three-day bins. **b**, The daily frequency of the predominant lineages and other

lineages in the seven countries (South Africa, Australia, Denmark, Germany, Israel, the UK and the USA) per day are shown. Unlike Fig. 1a and Extended Data Fig. 1a, the frequency of viral lineages other than the top five (for Australia, Denmark, Germany, Israel, and the UK) or six (for the USA) lineages are included. **c**, Comparison of observed and predicted counts of each viral lineage in each time bin. The results in the seven countries indicated are shown. The coefficient of determination ($R^2$) and the line y = x are shown. Each dot indicates the result of each viral lineage at each time point.

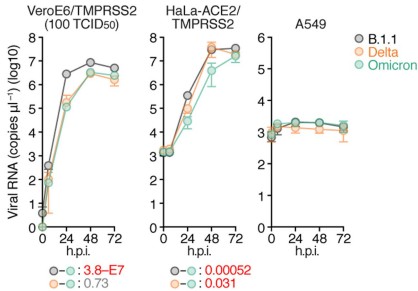

**Extended Data Fig. 2 | Growth of Omicron, Delta and B.1.1 in different cells.**
A D614G-bearing B.1.1 virus, Delta and Omicron [100 $TCID_{50}$ (m.o.i. = 0.01) for
VeroE6/TMPRSS2 cells, 1,000 $TCID_{50}$ (m.o.i. = 0.1) for HeLa-ACE2/TMPRSS2
cells and A549 cells] were inoculated into cells, and the viral RNA copy number
in the supernatant was quantified by RT–qPCR. Assays were performed in
quadruplicate. Data are mean ± s.d. In the data of VeroE6/TMPRSS2 cells (left)
and HeLa-ACE2/TMPRSS2 cells (middle), statistically significant differences
versus B.1.1 and Delta through time points were determined by multiple
regression. FWERs calculated using the Holm method are indicated in the
figures.

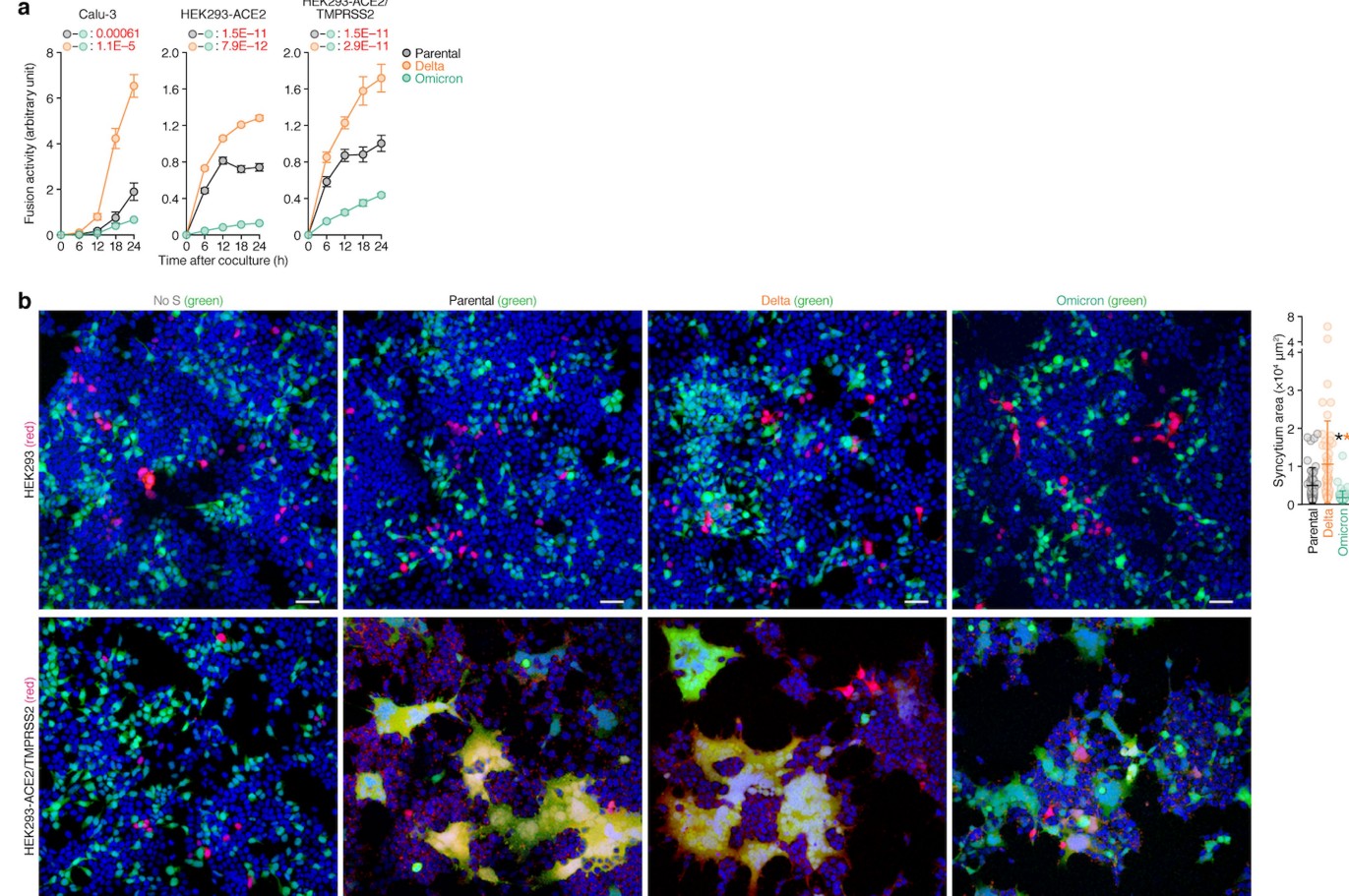

**Extended Data Fig. 3 | Cell–cell fusion mediated by the SARS-CoV-2 S protein. a**, SARS-CoV-2 S-based fusion assay. Effector cells (S-expressing cells) and target cells (Calu-3 cells, HEK293-ACE2 cells and HEK293-ACE2/TMPRSS2 cells) were prepared, and the fusion activity was measured as described in the Methods. Assays were performed in quadruplicate, and fusion activity (arbitrary units) is shown. **b**, Coculture of S-expressing cells with HEK293-ACE2/TMPRSS2 cells. Left, representative images of S-expressing cells (green) cocultured with HEK293 cells (red, top) or HEK293-ACE2/TMPRSS2 cells (red, bottom). Nuclei were stained with Hoechst33342 (blue). Scale bars, 50 μm. Right, the size distributions of syncytia (yellow) in the

HEK293-ACE2/TMPRSS2 cultures cocultured with the cells expressing the parental D614G S (50 yellow syncytia), Delta S (54 yellow syncytia) or Omicron S (58 yellow syncytia). Data are mean ± s.d. In **a**, statistically significant differences versus B.1.1 and Delta through time points were determined by multiple regression. FWERs calculated using the Holm method are indicated in the figures. In **b**, each dot indicates the result from an individual yellow syncytium. Statistically significant differences (\**P* < 0.05) versus B.1.1 (a black asterisk) and Delta (an orange asterisk) were determined by two-sided Mann–Whitney *U*-test.

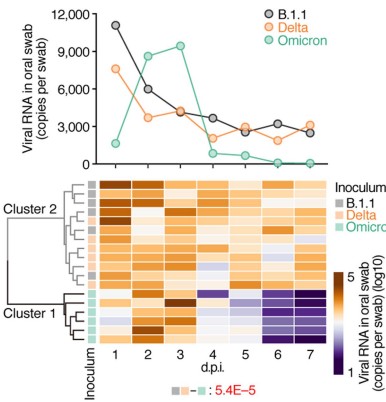

**Extended Data Fig. 4 | Dynamics of viral RNA load in oral swabs of infected hamsters.** The mean of viral RNA load in the oral swab (copies per swab) among infected hamsters at each time point is shown by a line plot in a linear scale (top), and the value in each infected hamster is shown by a heat map in a log scale (bottom). The result of the hierarchical clustering analysis is shown on the left of the heat map. The association between the clustering result and Omicron-infected hamsters was examined by two-sided Fisher's exact test, and the $P$ value is indicated in the figure.

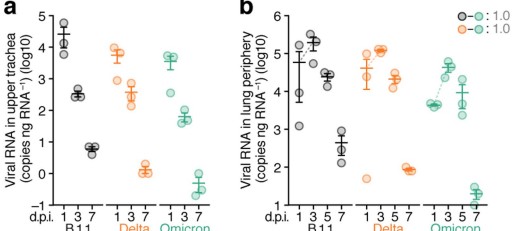

**Extended Data Fig. 5 | Quantification of viral RNA.** Syrian hamsters were intranasally inoculated with B.1.1 (n = 3), Delta (n = 3) and Omicron (n = 3). Viral RNA levels in the upper trachea (**a**) and lung periphery (**b**) were quantified by RT–qPCR. Data are mean ± s.e.m., and each dot indicates the result from an individual hamster. In **b**, the broken lines indicate the slopes of the average values between 1 d.p.i. and 3 d.p.i. Statistically significant differences of the slopes were determined by a likelihood-ratio test comparing the models with or without the interaction term of time point and inoculum. FWERs calculated using the Holm method are indicated in the figures.

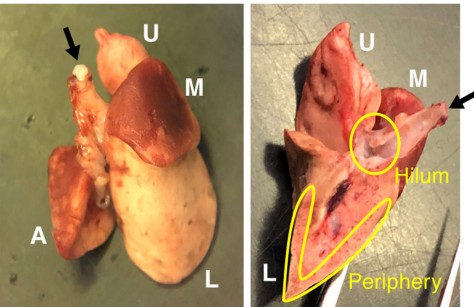

**Extended Data Fig. 6 | Analysed regions of the lung.** The entire lung (left) and a coronal section of the right lung and its cut surface (right) are shown. In the left panel, four lung lobes, the upper (anterior/cranial) lobe (U), milled lobe (M), lower (posterior/caudal) lobe (L) and accessory lobe (A), are respectively indicated. Arrow indicates the main bronchus. The hilum and periphery of the lung, which were used for the viral RNA quantification and titration (Fig. 4c and Extended Data Fig. 5b), are also indicated in yellow.

B.1.1

Delta

Omicron

19,368/53,268 (36.4%)
9,235/50,357 (18.3%)
5,421/48,323 (11.2%)

4,900/51,669 (9.5%)
4,713/53,516 (8.8%)
9,507/49,104 (19.4%)

39,561/55,806 (70.9%)
25,194/44,357 (56.8%)
14,018/44,337 (31.6%)

**Extended Data Fig. 7 | Morphometrical analysis of N-protein-positive bronchioles.** All four lobes of the right lung of infected hamsters (n = 3 for each virus) at 3 d.p.i. were immunohistochemically stained with anti-SARS-CoV-2 N monoclonal antibody. The circumference of all bronchioles (less than 500 μm diameter) is delineated in blue, and the positivity of N protein in bronchiole is indicated by magenta. Each length is indicated in the lower left of the panel by each colour. The number in parenthesis indicates the percentage of the N-positive bronchioles in the circumference of all bronchioles. The summarized result is shown in the right panel of Fig. 4d.

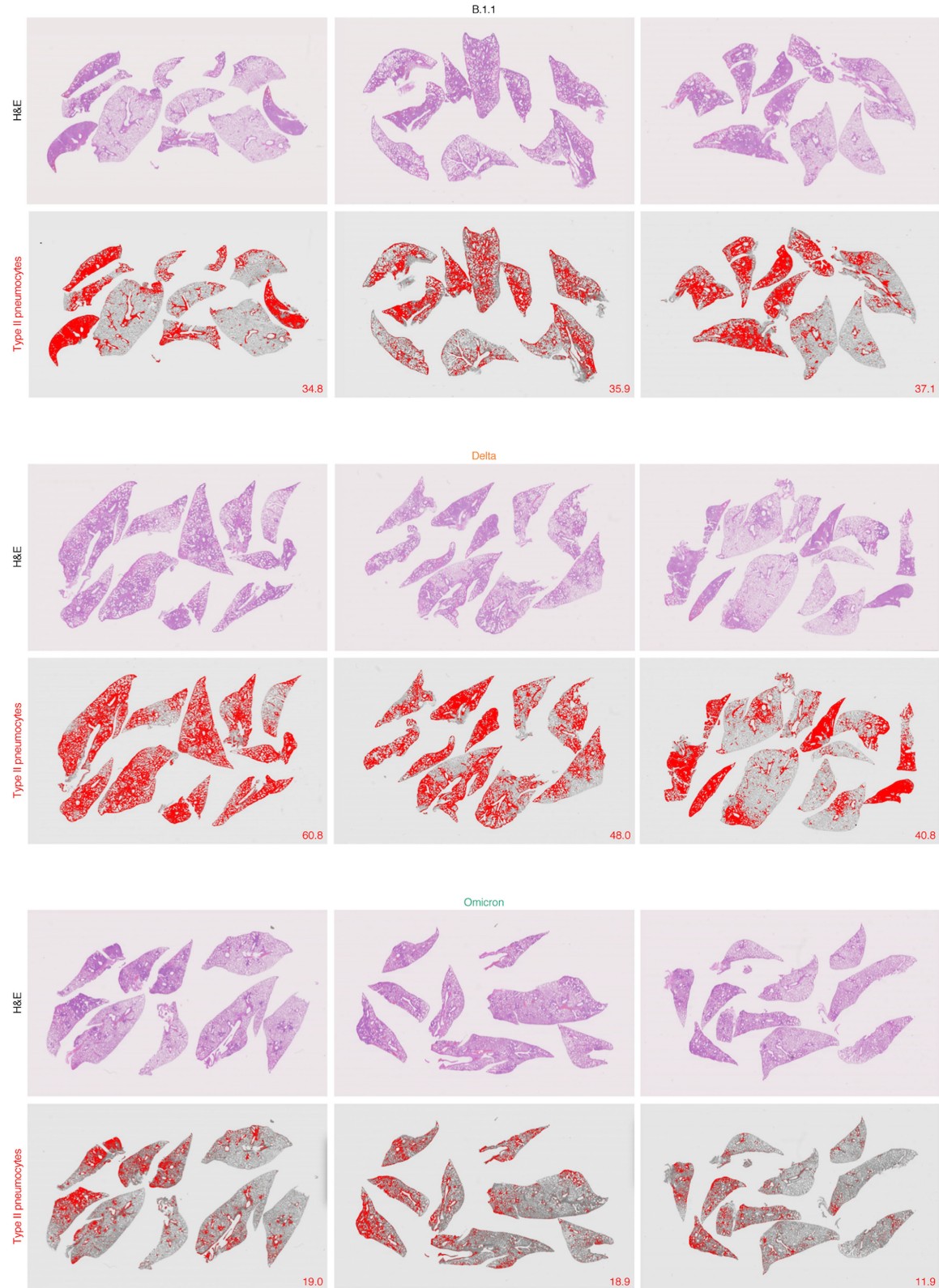

**Extended Data Fig. 8 | Type II pneumocytes in the lungs of infected hamsters.** Lung lobes of the hamsters infected with B.1.1 (top, n = 3), Delta (middle, n = 3), and Omicron (bottom, n = 3) at 5 d.p.i. In each panel, H&E staining (top) and the digitalized inflammation area (bottom, indicated in red) are shown. The number in the panel of the digitalized inflammation area indicates the percentage of the section represented by the indicated area (that is, the area indicated with red colour per the total area of the lung lobes). Note that the panels in the middle column are identical to those shown in Fig. 5a.

# Reporting Summary

## Statistics

For all statistical analyses, confirm that the following items are present in the figure legend, table legend, main text, or Methods section.

| n/a | Confirmed | |
|---|---|---|
| ☐ | ☒ | The exact sample size ($n$) for each experimental group/condition, given as a discrete number and unit of measurement |
| ☐ | ☒ | A statement on whether measurements were taken from distinct samples or whether the same sample was measured repeatedly |
| ☐ | ☒ | The statistical test(s) used AND whether they are one- or two-sided<br>*Only common tests should be described solely by name; describe more complex techniques in the Methods section.* |
| ☒ | ☐ | A description of all covariates tested |
| ☒ | ☐ | A description of any assumptions or corrections, such as tests of normality and adjustment for multiple comparisons |
| ☐ | ☒ | A full description of the statistical parameters including central tendency (e.g. means) or other basic estimates (e.g. regression coefficient) AND variation (e.g. standard deviation) or associated estimates of uncertainty (e.g. confidence intervals) |
| ☐ | ☒ | For null hypothesis testing, the test statistic (e.g. $F$, $t$, $r$) with confidence intervals, effect sizes, degrees of freedom and $P$ value noted<br>*Give P values as exact values whenever suitable.* |
| ☐ | ☒ | For Bayesian analysis, information on the choice of priors and Markov chain Monte Carlo settings |
| ☒ | ☐ | For hierarchical and complex designs, identification of the appropriate level for tests and full reporting of outcomes |
| ☒ | ☐ | Estimates of effect sizes (e.g. Cohen's $d$, Pearson's $r$), indicating how they were calculated |

*Our web collection on statistics for biologists contains articles on many of the points above.*

## Software and code

Policy information about availability of computer code

| | |
|---|---|
| Data collection | All-in-One Fluorescence Microscope BZ-X800 (Keyence)<br>QuantStudio 3 Real-Time PCR system (Thermo Fisher Scientific)<br>CFX Connect Real-Time PCR Detection system (Bio-Rad)<br>Eco Real-Time PCR System (Illumina)<br>qTOWER3 G Real-Time System (Analytik Jena)<br>7500 Real-Time PCR System (Thermo Fisher Scientific)<br>Centro XS3 LB960 (Berthhold Technologies)<br>FACS Canto II (BD Biosciences)<br>Amersham Imager 600 (GE Healthcare)<br>FinePointe Station and Review softwares v2.9.2.12849 (STARR)<br>MouseOx PLUS (STARR)<br>Autostainer Link 48 (Dako)<br>MiSeq (Illumina) |
| Data analysis | CmdStan v2.28.1 (https://mc-stan.org/users/interfaces/cmdstan)<br>cmdstanr v0.4.0 (https://mc-stan.org/cmdstanr/)<br>R v3.6.3 (https://www.r-project.org/)<br>Fiji software v2.2.0 (ImageJ)<br>Sequencher software v5.1 (Gene Codes Corporation)<br>FlowJo software v10.7.1 (BD Biosciences)<br>Image Studio Lite v5.2 (LI-COR Biosciences)<br>NDRscan3.2 software (Hamamatsu Photonics)<br>fastp v0.21.0 [Chen, et al. Bioinformatics 34, i884-i890 (2018). doi:10.1093/bioinformatics/bty560]<br>BWA-MEM v0.7.17 [Li & Durbin. Bioinformatics 26, 589-595 (2010). doi:10.1093/bioinformatics/btp698] |

SAMtools v1.9 [Danecek et al. Gigascience 10 (2021). doi:10.1093/gigascience/giab008]
snpEff v5.0e [Cingolani et al. Fly (Austin) 6, 80-92 (2012) doi:10.4161/fly.19695]
Excel software v16.16.8 (Microsoft)
Prism 9 software v9.1.1 (GraphPad Software)
The computational code to estimate the viral transmissibility (Fig. 1) is available in the GitHub repository (https://github.com/TheSatoLab/Estimation_of_transmissibility_of_each_viral_lineage).

For manuscripts utilizing custom algorithms or software that are central to the research but not yet described in published literature, software must be made available to editors and reviewers. We strongly encourage code deposition in a community repository (e.g. GitHub). See the Nature Portfolio guidelines for submitting code & software for further information.

## Data

Policy information about availability of data

All manuscripts must include a data availability statement. This statement should provide the following information, where applicable:
- Accession codes, unique identifiers, or web links for publicly available datasets
- A description of any restrictions on data availability
- For clinical datasets or third party data, please ensure that the statement adheres to our policy

The raw data of virus sequences analysed in this study are deposited in Gene Expression Omnibus (accession number: GSE192472). All databases/datasets used in this study are available from GISAID database (https://www.gisaid.org), Genbank database (https://www.ncbi.nlm.nih.gov/genbank/), Our World in Data (https://ourworldindata.org/covid-cases), or outbreak.info (https://outbreak.info). The accession numbers of viral sequences used in this study are listed in Method section.

# Field-specific reporting

Please select the one below that is the best fit for your research. If you are not sure, read the appropriate sections before making your selection.

☒ Life sciences ☐ Behavioural & social sciences ☐ Ecological, evolutionary & environmental sciences

For a reference copy of the document with all sections, see nature.com/documents/nr-reporting-summary-flat.pdf

# Life sciences study design

All studies must disclose on these points even when the disclosure is negative.

| | |
|---|---|
| Sample size | The sample sizes (n > 3) for cell culture experiments were chosen for applying statistical tests. The sample sizes (n > 3) for the hamster studies were chosen because they have previously been shown to be sufficient to evaluate a significant difference among groups (Belser et al., Nature, 2013; Zhang et al., Science, 2013; Imai et al., Nature Microbiology, 2020; Saito et al., Nature, 2021). |
| Data exclusions | In Fig. 3d, the SpO2 data of two infected hamsters at 5 d.p.i. were excluded because these hamsters were not restrained by the sedation. |
| Replication | In vitro experiments representative of at least 2 experiments with multiple samples per time point. In vivo experiments (hamster) utilized multiple animals per group per time point and were from more than single experiment. In vivo experiments were replicated and performed independently. All attempts at replication were successful. |
| Randomization | No method of randomization was used to determine how the animals were allocated to the experimental groups and processed in this study, because covariates (sex and age) were identical (male, 4 weeks old). For experiments other than animal studies, randomization is not applicable because homogenous materials (i.e., cell lines) were used. Primary human nasal epithelial cells were used in an experiment, but only one donor/batch was used. Therefore, randomization is not applicable. |
| Blinding | The number of investigators were limited, and most of experiments were performed in high-containment laboratories. Therefore, blinding was not carried out. |

# Reporting for specific materials, systems and methods

We require information from authors about some types of materials, experimental systems and methods used in many studies. Here, indicate whether each material, system or method listed is relevant to your study. If you are not sure if a list item applies to your research, read the appropriate section before selecting a response.

## Materials & experimental systems

| n/a | Involved in the study |
|---|---|
| ☐ | ☒ Antibodies |
| ☐ | ☒ Eukaryotic cell lines |
| ☒ | ☐ Palaeontology and archaeology |
| ☐ | ☒ Animals and other organisms |
| ☒ | ☐ Human research participants |
| ☒ | ☐ Clinical data |
| ☒ | ☐ Dual use research of concern |

## Methods

| n/a | Involved in the study |
|---|---|
| ☒ | ☐ ChIP-seq |
| ☐ | ☒ Flow cytometry |
| ☒ | ☐ MRI-based neuroimaging |

# Antibodies

| | |
|---|---|
| Antibodies used | Western blot:<br>mouse anti-SARS-CoV-2 S monoclonal antibody (clone 1A9, GeneTex, Cat# GTX632604, 1:10,000)<br>rabbit anti-SARS-CoV-2 N monoclonal antibody (clone HL344, GeneTex, Cat# GTX635679, 1:5,000)<br>rabbit anti-beta actin (ACTB) monoclonal antibody (clone 13E5, Cell Signalling, Cat# 4970, 1:5,000)<br>mouse anti-alpha tubulin (TUBA) monoclonal antibody (clone DM1A, Sigma-Aldrich, Cat# T9026, 1:10,000)<br>HRP-conjugated donkey anti-rabbit IgG polyclonal antibody (Jackson ImmunoResearch, Cat# 711-035-152, 1:10,000)<br>HRP-conjugated donkey anti-mouse IgG polyclonal antibody (Jackson ImmunoResearch, Cat# 715-035-150, 1:10,000)<br>For immunofluorescence staining:<br>rabbit anti-SARS-CoV-2 N polyclonal antibody (GeneTex, Cat# GTX135570, 1:1,000)<br>Alexa 488-conjugated anti-rabbit IgG antibody (Thermo Fisher Scientific, Cat# A-11008, 1:1,000)<br>Flow cytometry:<br>rabbit anti-SARS-CoV-2 S S1/S2 polyclonal antibody (Thermo Fisher Scientific, Cat# PA5-112048, 1:100)<br>mouse anti-SARS-CoV-2 S monoclonal antibody (clone 1A9, GeneTex, Cat# GTX632604, 1:100)<br>Normal rabbit IgG (SouthernBiotech, Cat# 0111-01, 1:100)<br>purified mouse IgG1 isotype control antibody (clone MG1-45, BioLegend, Cat# 401401, 1:100)<br>APC-conjugated goat anti-mouse or rabbit IgG polyclonal antibody (Jackson ImmunoResearch, Cat# 115-136-146, 1:50 or Cat# 111-136-144, 1:50)<br>IHC:<br>mouse anti-SARS-CoV-2 N monoclonal antibody (R&D systems, Clone 1035111, Cat# MAB10474-SP, 1:400) |
| Validation | Validation of all primary antibodies for the species and application was conducted by manufacturers prior to sale, and validation statements are available on the manufacturers' website. |

# Eukaryotic cell lines

Policy information about cell lines

| | |
|---|---|
| Cell line source(s) | HEK293 cells (a human embryonic kidney cell line; ATCC CRL-1573)<br>HEK293-ACE2/TMPRSS2 cells [HEK293 cells (ATCC CRL-1573) stably expressing human ACE2 and TMPRSS2; Motozono et al., Cell Host & Microbe, 2021)<br>A549 cells (a human lung epithelial cell line; ATCC CCL-185)<br>A549-ACE2 cells [A549 cells (ATCC CCL-185) stably expressing human ACE2; Motozono et al., Cell Host & Microbe, 2021]<br>Vero cells [an African green monkey (Chlorocebus sabaeus) kidney cell line; JCRB0111]<br>VeroE6/TMPRSS2 cells (JCRB1819)<br>Calu-3 cells (a human lung epithelial cell line; ATCC HTB-55)<br>Calu-3/DSP1-7 cells [Calu-3 cells (ATCC HTB-55) stably expressing DSP1-7; Yamamoto et al., Viruses, 2020)<br>HeLa-ACE2/TMPRSS2 cells [HeLa229 cells (JCRB9086) stably expressing human ACE2 and TMPRSS2; Kawase et al., Journal of Virology, 2012]<br>Primary human nasal epithelial cells (Cat# EP02, Batch# MP0010, Epithelix)<br>Vero cells, VeroE6/TMPRSS2 cells, and HeLa-ACE2/TMPRSS2 cells are commercially available at JCRB Cell Bank (https://cellbank.nibiohn.go.jp/english/). Primary human nasal epithelial cells were purchased from Epithelix. |
| Authentication | None of the cells used were authenticated. |
| Mycoplasma contamination | All cell lines were regularly tested for mycoplasma contamination by using PCR and were confirmed to be mycoplasma-free. |
| Commonly misidentified lines<br>(See ICLAC register) | No commonly misidentified cell lines were used. |

# Animals and other organisms

Policy information about studies involving animals; ARRIVE guidelines recommended for reporting animal research

| | |
|---|---|
| Laboratory animals | Syrian hamsters (male, 4 weeks old) were purchased from Japan SLC Inc. (Shizuoka, Japan). |
| Wild animals | No wild animal was used in this study. |

| Field-collected samples | No field collected sample was used in the study. |
|---|---|
| Ethics oversight | All experiments with hamsters were performed in accordance with the Science Council of Japan's Guidelines for Proper Conduct of Animal Experiments. The protocols were approved by the Institutional Animal Care and Use Committee of National University Corporation Hokkaido University (approval numbers 20-0123 and 20-0060). |

Note that full information on the approval of the study protocol must also be provided in the manuscript.

# Flow Cytometry

## Plots

Confirm that:

☒ The axis labels state the marker and fluorochrome used (e.g. CD4-FITC).

☒ The axis scales are clearly visible. Include numbers along axes only for bottom left plot of group (a 'group' is an analysis of identical markers).

☒ All plots are contour plots with outliers or pseudocolor plots.

☒ A numerical value for number of cells or percentage (with statistics) is provided.

## Methodology

| Sample preparation | HEK293 cells were cotransfected with S expression plasmids (400 ng) and pDSP1-7 (400 ng) using TransIT-LT1 (Takara, Cat# MIR2300). |
|---|---|
| Instrument | FACS Canto II instrument (BD Biosciences) |
| Software | FlowJo software v10.7.1 (BD Biosciences) |
| Cell population abundance | 10,000 cells gated in the FSC-A/SSC-A plot (Supplementary Fig. 1) were acquired for each condition. |
| Gating strategy | Live cell population was gated based on the FSC-A/SSC-A plot. The starting gating strategy is shown in Supplementary Fig. 1. Then, to define the boundary between "positive" and "negative" staining of the surface S protein, isotype control IgG was used instead of primary anti-S antibodies. Grey histograms in Fig. 2e and 2f indicate isotype controls of the assay. |

☒ Tick this box to confirm that a figure exemplifying the gating strategy is provided in the Supplementary Information.

