## [Peer Review File · Nature]

Manuscript Title: Attenuated fusogenicity and pathogenicity of SARS-CoV-2 Omicron variant

Redactions – unpublished data

Reviewer Comments & Author Rebuttals

Reviewer Reports on the Initial Version:

Referee #1:

The manuscript by Rigel Suzuki and colleagues combines modeling, in vitro and in vivo data to assess the danger posed by the omicron SARS-CoV-2 VOC that emerged recently. Using sequence data generated in South Africa and the UK, they first compare the transmissibility of omicron to other SARS-CoV-2 variants. Next, they perform an in vitro assessment of omicron, and show that it replicates less efficiently in several cell types than the delta variant and an ancestral strain, does not induce syncytium formation as efficiently as the delta variant, and that the S protein is cleaved less efficiently. Then, an experiment in hamsters is performed, again comparing omicron to the delta variant and an ancestral strain. These experiments show less weight loss, slightly lower viral loads in the lung periphery, and reduced histological lesions with omicron compared to the other two isolates. The authors conclude that omicron is less pathogenic than its' ancestor and the delta variant.

The experiments are performed well, but some clarification is needed for some of the experiments. I suggest the following improvements to the manuscript:

I am struggling a bit with the first section of the manuscript that estimates 'transmissibility' of the omicron variant. It is not clear to me why Fig. 1A&C and S1A&C both need to be included as they seem to show the same thing just on a slightly different scale (per week/per day datapoints). Nor is it explained why the modeling focuses on South Africa and the UK. Why is the UK included in this analysis? More importantly, I don't think the calculations provide a measure of transmissibility. They provide a measure of the speed of spread in the human population, but this is influenced by human factors, such as time since last wave of cases, seroprevalence, behavior (e.g. lockdown restrictions), and viral factors, such as escape from existing immunity and transmissibility. At the very least, this section needs to be rephrased. The second problem with this section is that it clashes a bit with the rest of manuscript, that focuses on virus replication and pathogenicity rather than on transmissibility. By first showing that omicron is more transmissible, one would subsequently expect experiments to address this transmissibility. But the opposite is true: the omicron isolate does not replicate as well, which makes the increased transmissibility data all the more puzzling.

The Methods indicate that virus stocks were sequenced before use, but I do not understand how to interpret the data in Supplemental File 2 as there are some differences compared to Extended Data Table 2. Please include clear statements on whether there were any changes detected (and at what %) in the stocks used in the experiments compared to the original isolates to ensure the absence of tissue culture adaptations that may affect the outcome of the experiments.

Genome copies are used in in vitro and in vivo experiments. These should be replaced with virus titers, as they are a much more important measure of virus replication. Moreover, there is a discrepancy between differences in viral loads in the hamsters (minimal differences between the different viruses) and the weight loss and histological lesions (large differences). This discrepancy

may be much less if infectious titers are shown.

The fusion experiments are very repetitive. I would suggest removing some of the data that are not quantitative, e.g. Extended data Figure 3 and maybe even Fig. 2b, c.

I can't figure out what the difference is between the experiment in Fig. 2g and 2h. Was this done in different cell types? And why would it need to be done in different cells? Also, the western blot bands look quite different for S and S2 between 2g and 2h (there are multiple bands in 2g but single bands in 2h). It is also not clear from the legend how the bands were quantified for the graphs in these panels. This should be explained more clearly.

Whole-body plethysmography was performed on infected hamsters. This usually results in many more parameters than PenH alone. Why are other parameters excluded?

Swabs (oral or nasal) over time should be included as a measure of what is going on in the upper respiratory tract. These were collected according to the Methods (lines 827-828) but I did not see the data.

The description of histological lesions is unconventional in text and figure. What is meant by 'inflammation with type II alveolar pneumocytes'? There are always type II pneumocytes in the alveoli, so first I thought the authors meant type II pneumocyte hyperplasia, but then later they mention 'large type II pneumocytes' which sounds to me more like that should indicate the hyperplasia. Please rephrase this section so it is clear what is meant.

In the discussion, it is proposed that the limited fusion capacity explains the reduced pathogenicity of the omicron isolate in hamsters. I think this is an overstatement. It is not clear whether this cell-cell fusion is also inhibited *in vivo* or whether virus is just not replicating to high titers and thus it is not clear that this is important for *in vivo* pathogenesis.

Minor corrections

Please check the manuscript and tables carefully for correct use of 'substitution' rather than 'mutation' when referring to amino acid changes.

Line 235-239. This sentence is very hard to follow. Please rephrase.

Extended Data Table 1. I am not sure this is useful. It would be more useful if sequences were compared to the ancestral strain used in this manuscript rather than Wuhan-1.

Extended Data Table 2 data can be removed as this is changing so fast that data will be obsolete at time of publishing, even if updated for a revised version of the manuscript.

Figure 3a-c. The use of light grey for uninfected makes these lines very difficult to see.

Fig. 3e. According to the legend this panel should have yellow arrows but I don't see those.

Extended data figure 5 is missing.

Referee #2:

This manuscript by Suzuki and co-workers describes the comparison of the replication kinetics, fusogenicity and pathogenicity of SARS-CoV-2 Omicron variant with previous variants of concern. This work is highly significant as it aims to determine if the mutations in the Omicron variant alter the virulence of an emerging virus in this pandemic. Figure 1 documents the rapid emergence and

transmission of Omicron in South Africa and in the United Kingdom, which is much more rapid than what occurred with previous variants. The authors document the differences in the growth kinetics, fusogenicity, plaque size and cleavage of the spike protein into the S1 and S2 subunits of Omicron, Delta and Parental SARS-CoV-2 in Figure 2. Importantly, they found that there is reduced cleavage of the Omicron spike to generate the S1 and S2 subunits during virus replication in cell culture. This phenotype of reduced cleavage of spike has previously been associated with delayed entry of MERS-CoV (Park et al., 2017 PNAS). They evaluated the pathogenicity of Omicron in the hamster model and found significantly reduced levels of virus replication and reduced pathology in the Omicron-infected hamsters compared to the Delta-infected hamsters (Figure 3). Overall, this study provides important information about the reduced pathogenesis of Omicron as compared to previous variants, and described some of the virological characteristics that correlate with the reduced pathogenesis (ie reduced cleavage of spike to generate S1 and S2 subunits). This study provides essential information for understanding how changes in the spike protein can alter the pathogenesis of emerging variants of SARS-CoV-2.

Comments for the authors' consideration:

1. The manuscript should be carefully edited to improve the clarity of the communication. A few specific examples of where the text should be modified are listed here:

Line 68-69: consider changing to Evaluating the proteolytic processing of spike revealed that the Omicron spike is less efficiently processed as compared to the Delta and parental spikes.

Line 100: be keen to evaluate?

Line 112-114: It is important to determine if the mutations in the S protein of Omicron....

Line 170: change it to we hypothesize

Line 273-274: is it immune resistance or that Omicron has evolved to evade the majority of the neutralizing antibodies generated to previous infections and vaccines?

Lines 304-306: please review for clarity

Lines 323-325: please review for clarity

2. Structural information on the Omicron spike is starting to become available (on BioRxiv) so the discussion could include comments on how the Omicron structure may allow efficient use of the ACE2 receptor but also have less efficient processing to generate S1 and S2 subunits.

3. Previous studies with MERS-CoV evaluated the role of cleavage of the spike on virus entry and could be included in the discussion (see Park et al., 2016 PNAS PMID: PMC5086990 and the references within this study).

Referee #3:

In this manuscript, Suzuki et al. perform various analyses to quantify characteristics of SARS-CoV-2 Variant of Concern Omicron. Here, given my expertise, I will only comment on their statistical modeling analyses, which estimated that Omicron is >3.0-fold more transmissible than Delta in South Africa and >5.6-fold more transmissible than this earlier VOC in the UK.

In the results section lines 119-133 and then also elsewhere (e.g. around line 318 in discussion), the authors repeatedly mention that omicron has higher transmissibility. More care should be taken to clarify that faster growth (at the population level) does not necessarily mean that Omicron is intrinsically more transmissible (that is, has a higher basic reproduction number R_0) than Delta. It could very well be that Omicron's effective reproductive number (R_e) is currently higher than that of Delta because of its ability to reinfect individuals previously infected with Delta. So, Omicron's increasing frequency in South Africa and elsewhere does not necessarily mean it is intrinsically more transmissible than Delta.

Regarding the multinomial logistic model described on lines 581-591: is this the same model that was used by Volz et al. (2020) Cell (10.1016/j.cell.2020.11.020)? If so, reference to this paper should be made. If there are differences from this approach, those should be highlighted. Also, and critically, while I understand the approach up to line 591, I don't understand how the relative growth rate per generation was quantified (lines 592-597). Specifically, in the equation at around line 595, are both γ and w 5.5 days (and therefore cancel one another?) Where is d in this equation?

Other estimates of Omicron's relative growth rate have been posted on preprint servers (e.g., Pulliam et al.). These I feel should be incorporated into the discussion section and compared against the estimates presented here.

Referee #4:

This a short, but solid study with respect to the infection and pathogenicity properties of Omicron, given the short time-frame involved.

At this point in time I am not convinced that Figure 1 really adds anything to the study

The authors need to be clearer in Fig 2 on what the noted statistical analysis is comparing (also with the asterix, especially for the Vero/TMRSS2 cells).

The authors also need to define better what the arbitrary units are for fusion activity.

In general, panels in Fig 2 need to be bigger, especially the Western blots – also please explain the band above S0 in parental (panel g).

Please use a better description of "faintly" for the Omicron syncytia

The hamster studies appear to be well done

Author Rebuttals to Initial Comments:

Referee #1 (Remarks to the Author)

The manuscript by Rigel Suzuki and colleagues combines modeling, in vitro and in vivo data to assess the danger posed by the omicron SARS-CoV-2 VOC that emerged recently. Using sequence data generated in South Africa and the UK, they first compare the transmissibility of omicron to other SARS-CoV-2 variants. Next, they perform an in vitro assessment of omicron, and show that it replicates less efficiently in several cell types than the delta variant and an ancestral strain, does not induce syncytium formation as efficiently as the delta variant, and that the S protein is cleaved less efficiently. Then, an experiment in hamsters is performed, again comparing omicron to the delta variant and an ancestral strain. These experiments show less weight loss, slightly lower viral loads in the lung periphery, and reduced histological lesions with omicron compared to the other two isolates. The authors conclude that omicron is less pathogenic than its' ancestor and the delta variant.

The experiments are performed well, but some clarification is needed for some of the experiments. I suggest the following improvements to the manuscript:

Our reply:

First of all, we are happy to hear "*The experiments are performed well*". We also thank the referee for providing the suggestions to improve our study.

I am struggling a bit with the first section of the manuscript that estimates 'transmissibility' of the omicron variant. It is not clear to me why Fig. 1A&C and S1A&C both need to be included as they seem to show the same thing just on a slightly different scale (per week/per day datapoints).

Our reply:

Fig. 1a and 1c of original manuscript show the processed data used as the input data for our statistical model. In these input data (shown in **Fig. 1a and 1c** of original manuscript; **Fig. 1a and Extended Data Fig. 1a** of revised manuscript), the number of sequences in each viral lineage in each day was summed in three-day bins, and the viral lineages other than the five (or six in the USA in the revised manuscript) most predominant lineages were excluded. On the other hand, **Extended Data Fig. 1a and 1c** of original manuscript (**Extended Data Fig. 1c** of revised manuscript) show the raw data of the relative frequency of respective viral lineages per day, which include not only the five (or six in the USA in the revised manuscript) most predominant lineages in the country (used in our modelling analysis) but also the other lineages (shown as "Others" in **Extended Data Fig. 1c** of revised manuscript). To clarify these points, we added the descriptions in the revised manuscript (**lines 492-495, page 15; lines 1070-1074, page 30**).

Nor is it explained why the modeling focuses on South Africa and the UK. Why is the UK included in this analysis?

Our reply:

In the original manuscript, we included the UK data because the number of the sequence records of Omicron from the UK was clearly higher than those from the other countries (including South Africa) in the GISAID dataset we used (downloaded on December 20, 2021). However, after the initial submission, Omicron has spread worldwide, and >10,000 records of the Omicron from the countries other than South Africa and the UK were uploaded in the GISAID database. Therefore, in the revised manuscript, we updated the working GISAID dataset (downloaded on January 4, 2022) and included the seven countries with >1,500 Omicron sequences (Australia,

Denmark, Germany, Israel, South Africa, the UK, and the USA) in our analysis (**Fig. 1 and Extended Data Fig. 1**).

More importantly, I don't think the calculations provide a measure of transmissibility. They provide a measure of the speed of spread in the human population, but this is influenced by human factors, such as time since last wave of cases, seroprevalence, behavior (e.g. lockdown restrictions), and viral factors, such as escape from existing immunity and transmissibility. At the very least, this section needs to be rephrased.

Our reply:

Thank you for the important comment. We agree that the term “transmissibility” is not appropriate to interpret the results of our statistical modelling analysis. In the revised manuscript, we used the term “relative growth rate per generation”, which was used in the earlier works [Vohringer et al., Nature, 2021. (PMID 34649268); Davies et al., Science, 2021 (PMID 33658326); Obermeyer et al., medRxiv, 2021 (doi: 10.1101/2021.09.07.21263228)] employing statistical approach similar to ours, instead of the term “transmissibility”. To clarify this, we referred these papers and a preprint in the revised manuscript.

The second problem with this section is that it clashes a bit with the rest of manuscript, that focuses on virus replication and pathogenicity rather than on transmissibility. By first showing that omicron is more transmissible, one would subsequently expect experiments to address this transmissibility. But the opposite is true: the omicron isolate does not replicate as well, which makes the increased transmissibility data all the more puzzling.

Our reply:

We agree that the most important and impressive result of the present work is the less pathogenicity of Omicron in a hamster model rather than its higher transmissibility. However, we believe that the results suggesting a higher growth rate of Omicron in the human population (**Fig. 1 and Extended Data Fig. 1**) should be included in the manuscript—if we only show the data suggesting the attenuated pathogenicity of Omicron, certain readers may misunderstand the message of our study and underestimate the social risks of Omicron. Our results in a hamster model

suggest the attenuated pathogenicity of Omicron, however, these results do not necessarily mean the risk of Omicron for global health is relatively low. This is because viral pathogenicity has a linear effect on the increase in hospital admissions, severe cases, and deaths, whereas viral transmissibility has an exponential effect on these factors. We carefully discuss this point in the last paragraph of the Discussion section in the revised manuscript (lines 354-364, pages 10-11). To clearly show the message and conclusion of our study, we think the data shown in **Fig. 1 and Extended Data Fig. 1** are essential.

Regarding "*But the opposite is true: the omicron isolate does not replicate as well, which makes the increased transmissibility data all the more puzzling.*"; we agree with this comment. However, as mentioned in the Discussion section (lines 317-324, page 10), the growth of Delta, which surpassed other variants and was the dominant causative agent of the SARS-CoV-2 pandemic in the last year, in both cell cultures and a hamster model was not higher than that of an early pandemic SARS-CoV-2, and these observations are consistent with our previous study [Saito et al., *Nature*, 2021 (PMID 34823256)]. Therefore, our data suggest that the growth capacity of SARS-CoV-2 in cell cultures and animal models does not necessarily reflect human-to-human transmissibility. Moreover, in the revised manuscript, we provided the data suggesting the reason why Omicron is highly transmissible (**Fig. 3e, 4c and 4d** of revised manuscript). We hope our new data and explanation are acceptable by the referee.

The Methods indicate that virus stocks were sequenced before use, but I do not understand how to interpret the data in Supplemental File 2 as there are some differences compared to Extended Data Table 2. Please include clear statements on whether there were any changes detected (and at what %) in the stocks used in the experiments compared to the original isolates to ensure the absence of tissue culture adaptations that may affect the outcome of the experiments.

Our reply:

We apologize for the confusing data presentation in **Extended Data Table 1** (probably "Extended Data Table 2" mentioned by the referee would be "Extended Data Table 1") and **Supplemental Table 2** of original manuscript. The purpose of showing **Extended Data Table 1** is to describe the number of mutations in Delta and Omicron (lines 99-107, page 4), whereas that of showing **Supplemental Table 2** is to validate the sequences of working viruses, as indicated by the referee. According to the referee's suggestion, we modified **Supplemental Table 2** in the revised manuscript; we summarized the mutations that were not detected in the original

isolate stocks but in the working virus stocks used in the analysis. For instance, in the working virus of the B.1.1 isolate (EPI_ISL_479681), no additional mutation was detected compared to the original stock. In the working viruses of Delta (EPI_ISL_2378732) and Omicron (EPI_ISL_7418017), although some nonsynonymous substitutions (three for Delta and two for Omicron) were detected, these substitutions are not included in the spike protein.

Because **Extended Data Table 1** is essential to compare the number of mutations between Delta and Omicron, we left this data in the revised manuscript.

Genome copies are used in in vitro and in vivo experiments. These should be replaced with virus titers, as they are a much more important measure of virus replication. Moreover, there is a discrepancy between differences in viral loads in the hamsters (minimal differences between the different viruses) and the weight loss and histological lesions (large differences). This discrepancy may be much less if infectious titers are shown.

Our reply:

Discrepancy between viral loads in the hamsters (minimal differences between the different viruses) and the weight loss and histological lesions (large differences):

The time-course analysis of the viral antigen in lung (i.e., the IHC of viral N protein in lung, shown in **Fig. 3e** of the original manuscript; **Fig. 4b** of the revised manuscript) showed a clear difference between Omicron and Delta. Therefore, the difference of the pathological observations between Omicron and Delta can be explained by the difference of the efficacy of virus spread in lung, which is shown in **Fig. 4b**. In the revised manuscript, we added some data of the difference of viral spread that can explain the difference of viral pathogenicity (e.g., **Fig. 3e and 4**). Based on the data, we blushed up the Discussion section. We hope the revised manuscript is acceptable by the referee.

Virus titers in in vivo experiments:

To satisfy the referee's concern, we quantified viral titers (TCID₅₀) of the samples obtained from the lung hilum of infected hamsters, and the data are shown in **Fig. 4c** (bottom) of revised manuscript. We think this result is very supportive for our conclusion. Thank you very much for providing important suggestion.

Virus titers in in vitro experiments:

We understand this suggestion. However, because of the following reasons, we think re-measuring virus replication by virus titers would not be essential in our study:

1. As mentioned in the text, the efficacy of virus growth in in vitro cell culture does not explain the growth capacity and pathogenicity of SARS-CoV-2 in vivo. Rather, the efficacy of S cleavage and viral fusogenicity is closely associated with viral pathogenicity. This is one of the main messages of our study, and this is already shown.
2. In previous studies [e.g., Thomson et al., *Cell*, 2021 (PMID 33621484); Chu et al., *Lancet Microbe*, 2020 (PMID 32835326); Puray-Chavez et al., *Cell Rep*, 2021 (PMID 34214467); Higuchi et al., *Nat Commun*, 2021 (PMID 34155214)] and our recent study [Saito et al., *Nature*, 2021 (PMID 34823256)], virus replication was monitored by viral RNA in culture supernatant.
3. More importantly, as shown in **Fig. 2** of original manuscript, the cytopathic effect (CPE) of Omicron is much less than those of Delta and the B.1.1 virus. Our observations are well consistent with those reported in the other preprints, and we think it is evident that Omicron exhibits less CPE in cell cultures. Therefore, in our experience, measuring virus replication by CPE (i.e., TCID or p.f.u.) may underestimate the level of infectious Omicron in the specimen, and measuring it by viral RNA (as shown in our current study) would be more precise and straightforward.

The fusion experiments are very repetitive. I would suggest removing some of the data that are not quantitative, e.g. Extended data Figure 3 and maybe even Fig. 2b, c.

Our reply:

According to the suggestion, we removed **Extended Data Fig. 3** of the original manuscript. However, as recognized by the referee 2 (below), we think the association between viral fusogenicity and viral pathogenicity is one of the main messages of our study. To clearly show the difference of viral fusogenicity, we thought **Fig. 2b and 2c** are important for the readers and left them in the revised manuscript.

I can't figure out what the difference is between the experiment in Fig. 2g and 2h. Was this done in different cell types? And why would it need to be done in different cells? Also, the western blot bands look quite different for S and S2 between 2g and 2h (there are multiple bands in 2g but single bands in 2h). It is also not clear from the legend how the bands were quantified for the graphs in these panels. This should be explained more clearly.

Our reply:

Sorry for the confusion. **Fig. 2g** of the original manuscript (**Fig. 2h** of revised manuscript) is the blot of "S-expressing cells", while **Fig. 2h** of the original manuscript (**Fig. 2i** of revised manuscript) is the blot of "SARS-CoV-2-infected cells". This was mentioned in the Result section (lines 168-174, page 6), figure legend (lines 512-514, page 15), and the Method section (lines 843-846, page 24) of revised manuscript.

Whole-body plethysmography was performed on infected hamsters. This usually results in many more parameters than PenH alone. Why are other parameters excluded?

Our reply:

Thank you very much for the important suggestion. According to the referee's comment, we additionally assessed the other parameters. In the revised manuscript, we additionally showed the R_{pef} value in **Fig. 3c** of revised manuscript. The R_{pef} value nicely showed the difference between Omicron and the other viruses (B.1.1 and Delta). Thank you again for the critical suggestion.

Swabs (oral or nasal) over time should be included as a measure of what is going on in the upper respiratory tract. These were collected according to the Methods (lines 827-828) but I did not see the data.

Our reply:

Thank you very much for the important comment. According to the referee's suggestion, we quantified viral RNA load in the oral swab over time. The data are shown in **Fig. 3e** of revised manuscript. We think this results is very important to understand the dynamics of viral growth in Omicron. Based on the new data, we modified the manuscript. Thank you very much for providing important suggestion.

This may relate to the referee's comment above ("**Genome copies are...**"). We have tried quantifying viral titer in the oral swab. However, as the referee may know, the amount of the oral swab obtained from an infected hamster is very tiny. Also, as the referee indicated, we have already collected the oral swab during preparing the original manuscript. But as mentioned above, because the amount of the oral swab is very small, the collected sample was stored in one vial per timepoint per infected hamster. Then, during the evaluation of the manuscript (i.e., before receiving the referees' comments), we have already started quantifying viral RNA load by thawing the vials stored at -80°C . We re-stored the remaining samples and then tried quantified viral titer by re-thawing, according to the referee's suggestion. However, because 1) the initial amount was too low; and 2) the sample was frozen-and-thawed repeatedly, we unfortunately could not quantify viral titer in the oral swab.

The description of histological lesions is unconventional in text and figure. What is meant by 'inflammation with type II alveolar pneumocytes'? There are always type II pneumocytes in the alveoli, so first I thought the authors meant type II pneumocyte hyperplasia, but then later they mention 'large type II pneumocytes' which sounds to me more like that should indicate the hyperplasia. Please rephrase this section so it is clear what is meant.

Our reply:

As the referee commented, proliferation of type II pneumocytes is hyperplasia. According to the referee's suggestion, we rephrased the word "*large type II pneumocytes*" with "*large type II pneumocyte hyperplasia*" (e.g., lines 274, 278, 298, 299, page 9 in the Result section; pages 26-27 in the Method section). Additionally, we brushed up the sentences describing histological observations in the revised manuscript.

Regarding "*large type II pneumocytes*"; in our previous study [Saito et al., *Nature*, 2021 (PMID 34823256)], we observed prominently large pneumocytes exhibiting $>10\text{-}\mu\text{m}$ -diameter nucleus in the lungs of SARS-CoV-2-infected (particularly Delta-infected) hamsters. We explained it in the Method section of the revised manuscript (lines 944-948, pages 26-27): *The "large type II pneumocytes"*

are the hyperplasia of type II pneumocytes exhibiting more than 10- μ m-diameter nucleus. We described "large type II pneumocytes" as one of the remarkable histopathological features reacting SARS-CoV-2 infection in our previous study [Saito et al., Nature, 2021 (PMID 34823256)].

In the discussion, it is proposed that the limited fusion capacity explains the reduced pathogenicity of the omicron isolate in hamsters. I think this is an overstatement. It is not clear whether this cell-cell fusion is also inhibited in vivo or whether virus is just not replicating to high titers and thus it is not clear that this is important for in vivo pathogenesis.

Our reply:

According to the referee's suggestion, we toned down the sentences in the Discussion section. However, again, we think the association between viral fusogenicity and viral pathogenicity is one of the main messages of our study and the referee 2 understands its importance. We referred additional references to support this possibility (lines 335-353, page 10).

Minor corrections

Please check the manuscript and tables carefully for correct use of 'substitution' rather than 'mutation' when referring to amino acid changes.

Our reply: We fixed it in the revised manuscript (e.g, lines 79, 80, page 4; **Extended Data Table 1**).

Line 235-239. This sentence is very hard to follow. Please rephrase.

Our reply: Sorry for our poor sentences. The revised manuscript was proofread by the expertized editors in Nature Publishing group service. We hope the sentence was improved.

Extended Data Table 1. I am not sure this is useful. It would be more useful if sequences were compared to the ancestral strain used in this manuscript rather than Wuhan-1.

Our reply: As mentioned above, the purpose of showing **Extended Data Table 1** is to describe the number of mutations in Delta and Omicron compared with the original SARS-CoV-2 (Wuhan-1) (lines 99-107, page 4), whereas that of showing **Supplemental Table 2** is to validate the sequences of working viruses, as indicated by the referee. According to the referee's suggestion, we modified **Supplemental Table 2** of revised manuscript; we summarized the mutations that were not detected in the original isolate stocks but in the working virus stocks used in the analysis.

Extended Data Table 2 data can be removed as this is changing so fast that data will be obsolete at time of publishing, even if updated for a revised version of the manuscript.

Our reply: According to the suggestion, we removed **Extended Data Table 2** of original manuscript from the revision.

Figure 3a-c. The use of light grey for uninfected makes these lines very difficult to see.

Our reply: Sorry for our inconvenience. We modified the colors of the figure panels of revised manuscript.

Fig. 3e. According to the legend this panel should have yellow arrows but I don't see those.

Our reply: The arrows are shown in grey in **Fig. 4b** of revised manuscript (**Fig. 3e** of original manuscript).

Extended data figure 5 is missing.

Our reply: Sorry again for our mistake. It is included in the revised manuscript as **Extended Data Fig. 6**.

Referee #2 (Remarks to the Author)

This manuscript by Suzuki and co-workers describes the comparison of the replication kinetics, fusogenicity and pathogenicity of SARS-CoV-2 Omicron variant with previous variants of concern. This work is highly significant as it aims to determine if the mutations in the Omicron variant alter the virulence of an emerging virus in this pandemic. Figure 1 documents the rapid emergence and transmission of Omicron in South Africa and in the United Kingdom, which is much more rapid than what occurred with previous variants. The authors document the differences in the growth kinetics, fusogenicity, plaque size and cleavage of the spike protein into the S1 and S2 subunits of Omicron, Delta and Parental SARS-CoV-2 in Figure 2. Importantly, they found that there is reduced cleavage of the Omicron spike to generate the S1 and S2 subunits during virus replication in cell culture. This phenotype of reduced cleavage of spike has previously been associated with delayed entry of MERS-CoV (Park et al., 2017 PNAS). They evaluated the pathogenicity of Omicron in the hamster model and found significantly reduced levels of virus replication and reduced pathology in the Omicron-infected hamsters compared to the Delta-infected hamsters (Figure 3). Overall, this study provides important information about the reduced pathogenesis of Omicron as compared to previous variants, and described some of the virological characteristics that correlate with the reduced pathogenesis (ie reduced cleavage of spike to generate S1 and S2 subunits). This study provides essential information for understanding how changes in the spike protein can alter the pathogenesis of emerging variants of SARS-CoV-2.

Our reply:

First of all, we are very happy to hear *"This work is highly significant as it aims to determine if the mutations in the Omicron variant alter the virulence of an emerging virus in this pandemic"* and *"This study provides essential information for understanding how changes in the spike protein can alter the pathogenesis of emerging variants of SARS-CoV-2"*. Moreover, We also thank the referee for providing important suggestions.

Comments for the authors' consideration:

1. The manuscript should be carefully edited to improve the clarity of the communication. A few specific examples of where the text should be modified are listed here:

Line 68-69: consider changing to Evaluating the proteolytic processing of spike revealed that the Omicron spike is less efficiently processed as compared to the Delta and parental spikes.

Line 100: be keen to evaluate?

Line 112-114: It is important to determine if the mutations in the S protein of Omicron....

Line 170: change it to we hypothesize

Line 273-274: is it immune resistance or that Omicron has evolved to evade the majority of the neutralizing antibodies generated to previous infections and vaccines?

Lines 304-306: please review for clarity

Lines 323-325: please review for clarity

Our reply:

Sorry for our poor sentences. We modified the revised manuscript to the best of our abilities. Moreover, the revised manuscript was proofread by the expertized editors in the Nature Publishing Group service. We hope the sentences in revised manuscript were improved.

2. Structural information on the Omicron spike is starting to become available (on BioRxiv) so the discussion could include comments on how the Omicron structure may allow efficient use of the ACE2 receptor but also have less efficient processing to generate S1 and S2 subunits.

Our reply:

Thank you for the suggestion. We agree that understanding how Omicron S exhibits less efficient processing compared to Delta S and parental S is important. However, in our understanding, structural information of the Omicron S cannot help understanding it, and we do not have any answers explaining it. This will be one of the main topics on Omicron in the near future. Instead of including our comments on how the Omicron structure may allow efficient use of the ACE2 receptor but also have less efficient processing to generate S1 and S2 subunits, we referred a preprint reporting Omicron S structure and mentioned our understanding in the discussion

section of revised manuscript it as follows (lines 335-337, page 10): *Although the crystal structure of Omicron S is revealed [Mannar et al., bioRxiv, 2021 (doi: 10.1101/2021.1112.1119.473380)], the molecular and structural mechanisms of how Omicron S is resistant to the furin-mediated cleavage remains unclear.*

3. Previous studies with MERS-CoV evaluated the role of cleavage of the spike on virus entry and could be included in the discussion (see Park et al., 2016 PNAS PMID: PMC5086990 and the references within this study).

Our reply:

We would appreciate the supportive suggestion. We carefully read the paper indicated by the referee [Park et al., PNAS, 2016 (PMID 27791014)]. The relevance the S cleavage of MERS-CoV and its lung infection is interesting. However, unfortunately, its association with viral pathogenicity (using animal models) was not shown in this study. Therefore, we did not include this paper in the revised manuscript. However, the referee's comment was thought-provoking and we added a paragraph regarding the association between the S cleavage and viral pathogenicity in the Discussion section of revised manuscript (lines 335-353, page 10). Thank you very much for helpful suggestion.

Referee #3 (Remarks to the Author)

In this manuscript, Suzuki et al. perform various analyses to quantify characteristics of SARS-CoV-2 Variant of Concern Omicron. Here, given my expertise, I will only comment on their statistical modeling analyses, which estimated that Omicron is >3.0-fold more transmissible than Delta in South Africa and >5.6-fold more transmissible than this earlier VOC in the UK.

In the results section lines 119-133 and then also elsewhere (e.g. around line 318 in discussion), the authors repeatedly mention that omicron has higher transmissibility. More care should be taken to clarify that faster growth (at the population level) does not necessarily mean that Omicron is intrinsically more transmissible (that is, has a higher basic reproduction number R_0) than Delta. It could very well be that Omicron's effective reproductive number (R_e or R_{eff}) is currently higher than that of Delta because of its ability to reinfect individuals previously infected with Delta. So, Omicron's increasing frequency in South Africa and elsewhere does not necessarily mean it is intrinsically more transmissible than Delta.

Our reply:

Thank you very much for the important comment. We agree that the term "transmissibility" (or basic reproduction number R_0) is not appropriate to interpret the results of our statistical modelling analysis. In the revised manuscript, we used the term "relative growth rate per generation", which was used in the earlier works [Vohringer et al., *Nature*, 2021. (PMID 34649268); Davies et al., *Science*, 2021 (PMID 33658326); Obermeyer et al., *medRxiv*, 2021 (doi: 10.1101/2021.09.07.21263228)] employing statistical approach similar to ours, instead of the term "transmissibility".

Regarding the multinomial logistic model described on lines 581-591: is this the same model that was used by Volz et al. (2020) *Cell* (10.1016/j.cell.2020.11.020)?

Our reply:

We checked the Volz et al. paper indicated and found that this study did not use a multinomial logistic regression model, which is used in our study. Instead, Volz et al.

used a binomial logistic regression model. A binomial logistic regression model is used for a two-class comparison problem (e.g., the comparison between D614G vs. non-D614G), while a multinomial logistic regression model is used for a multi-class comparison problem (e.g., the comparisons among Alpha, Beta, Delta, and Omicron).

There are several previous studies [e.g., Vöhringer et al., Nature, 2021 (PMID 34649268) and Obermeyer et al., medRxiv, 2021 (doi: 10.1101/2021.09.07.21263228)] that used a multinomial logistic regression model to estimate the relative growth rates of SARS-CoV-2 lineages. However, the structure of the models used in these studies above are substantially different from ours: the model in Vöhringer et al. used a Dirichlet-multinomial distribution instead of a simple multinomial distribution. The model in Obermeyer et al. did not directly estimate the relative growth rate of each viral lineage. Instead, the model in Obermeyer et al. estimated the slopes for respective nonsynonymous substitutions, and the relative growth rate of a viral lineage is calculated according to the linear combination of the slopes of nonsynonymous substitutions present in the lineage. Importantly, although our statistical model is quite simple, our fitted model closely recapitulated the observed viral lineage dynamics in each country ($R^2 > 0.99$ in all the countries; **Extended Data Fig. 1d** of revised manuscript) in our dataset.

If so, reference to this paper should be made. If there are differences from this approach, those should be highlighted.

Our reply:

As mentioned above, we agree that previous studies (e.g., Volz et al.) used a similar (but not identical) statistical model to ours. Therefore, we referred to these studies [Vöhringer et al., Nature, 2021. (PMID 34649268); Davies et al., Science, 2021 (PMID 33658326); Obermeyer et al., medRxiv, 2021 (doi: 10.1101/2021.09.07.21263228)] in the revised manuscript.

Also, and critically, while I understand the approach up to line 591, I don't understand how the relative growth rate per generation was quantified (lines 592-597). Specifically, in the equation at around line 595, are both γ and w 5.5 days (and therefore cancel one another?) Where is d in this equation?

Our reply:

Sorry for the confusion. In the original manuscript, “d” meant “day”. To avoid confusion, we spelled out “d” as “day” in the revised manuscript (lines 642-643, page 19). Also, “ γ (gamma)” (fixed viral generation time) and “w” (time bin size) are set at 5.5 and 3 days, respectively. Therefore, “ γ (gamma)” and “w” do not cancel each other out.

Other estimates of Omicron’s relative growth rate have been posted on preprint servers (e.g., Pulliam et al.). These I feel should be incorporated into the discussion section and compared against the estimates presented here.

Our reply:

Thank you for the important suggestion. We agree that there are earlier works that estimate the relative growth rate of Omicron.

We carefully read the preprint indicated by the referee [Pulliam et al., 2021, MedRxiv (doi: 10.1101/2021.11.11.21266068)] and found that this work did not perform the estimation of the relative growth rate of Omicron. Instead, Pulliam et al. showed that the "reinfection risk" of SARS-CoV-2 in the epidemic surge driven by Omicron is higher than those driven by other SARS-CoV-2 lineages in South Africa. Because Omicron's "reinfection risk" is not the main scope of this study, we did not incorporate this preprint as a reference. Instead, in the revised manuscript, we referred one of the earliest works that estimated the relative growth rate of Omicron in South Africa and was published in a peer-reviewed journal [Nishiura et al., J Clin Med, 2022 (doi: 10.3390/jcm11010030)].

Referee #4 (Remarks to the Author):

This a short, but solid study with respect to the infection and pathogenicity properties of Omicron, given the short time-frame involved.

Our reply:

First of all, we are very happy to hear this is a "*solid study*". We thank the referee for providing important suggestions.

At this point in time I am not convinced that Figure 1 really adds anything to the study

Our reply:

Thank you for the comment. We know that some other studies showing higher transmissibility of Omicron in South Africa are already published as peer-reviewed papers and preprints. Therefore, to straighten the impact of transmissibility data, we additionally used the data obtained from six countries, Australia, Denmark, Germany, Israel, the UK and the USA, where >1,500 Omicron sequences reported at the time of writing this paper. These additional data are shown in **Fig. 1c and Extended Data Fig. 1** of revised manuscript. We believe these are novel.

Also, we agree that the transmissibility data shown in **Fig. 1** of the original manuscript may not be directly related to the other data. However, we believe that the results suggesting a higher growth rate of Omicron in the human population should be included in this study—if we only show the data suggesting the attenuated pathogenicity of Omicron, certain readers may misunderstand the message of our study and underestimate the social risks of Omicron. Our results in a hamster model suggest the attenuated pathogenicity of Omicron, however, these results do not necessarily mean the risk of Omicron for global health is relatively low. This is because viral pathogenicity has a linear effect on the increase in hospital admissions, severe cases, and deaths, whereas viral transmissibility has an exponential effect on these factors. We carefully discussed this point in the last paragraph of the Discussion section in the revised manuscript (**lines 354-364, pages 10-11**). To clearly show the message and conclusion of our study, we think showing the data of Omicron's higher transmissibility is important.

The authors need to be clearer in Fig 2 on what the noted statistical analysis is comparing (also with the asterix, especially for the Vero/TMRSS2 cells).

Our reply:

In the revised manuscript, we changed the statistical test applied to the time-course data (multiple regression analysis including experimental conditions as explanatory variables and timepoints as qualitative control variables), which would be easier to understand and further a better.

The authors also need to define better what the arbitrary units are for fusion activity.

Our reply:

This value was calculated as “luminescence (exhibited by cell-cell fusion) per the surface S MFI”. Therefore, this value has no unit (arbitrary). Since this presentation style has been accepted in our recent paper [Saito et al., *Nature*, 2021 (PMID 34823256)], we think this is acceptable. Because the word count of figure legend is limited, this is explained in the Method section of revised manuscript (lines 820-823, pages 23-24).

In general, panels in Fig 2 need to be bigger, especially the Western blots – also please explain the band above S0 in parental (panel g).

Our reply:

Fig. 2 in the revised manuscript was modified to the best of our ability. In particular, the Western blots (**Fig. 2g,h** of original manuscript; **Fig. 2h,i** of revised manuscript) were enlarged.

Regarding the extra band in the Western blot (**Fig. 2g** of original manuscript; **Fig. 2h** of revised manuscript); thank you very much for the critical notification. It was

an unspecific band detected only in this blot. To avoid misunderstanding, we replaced the blots (S/S2 and ACTB) with another ones in the revised manuscript.

Please use a better description of “faintly” for the Omicron syncytia

Our reply:

Fixed (“faintly” is replaced with “only weakly”; lines 147, page 5).

The hamster studies appear to be well done

Our reply:

Much appreciated. In the revised manuscript, we added more data addressing the viral growth of Omicron in vivo. We hope our revised manuscript is acceptable by the referee.

Reviewer Reports on the First Revision:

Referee #1:

The authors have done a great job incorporating the suggestions of the reviewers. My only remaining comment is that the authors forgot to replace the mention of ‘more transmissible’ in line 61.

Referee #3:

I think the revisions to this manuscript have improved the presented work, but I have several remaining concerns.

Conclusion that Omicron has higher transmissibility than Delta:

Reviewer 1 and reviewer 3 (me) both commented on the inappropriate conclusion by the authors, based on their analyses, that Omicron is “more transmissible” than Delta. The authors also state in their response letter (here in response to reviewer 1) that ‘We agree that the term ‘transmissibility’ is not appropriate to interpret the results of our statistical modelling analysis’. Despite this agreement, the manuscript still repeatedly states this conclusion, which is not explicitly supported by their analyses. The analyses indicate that Omicron is spreading more rapidly than Delta (that is, currently has a higher growth rate). But this could be because of Omicron’s ability to escape immunity, not intrinsically higher transmissibility of Omicron relative to Delta. To me, the

remaining text concluding higher transmissibility is a big concern, as a paper published in Nature has the potential to be heavily cited, and this conclusion is not supported by their analyses. Almost certainly, this would prompt letters to Nature expressing disagreement with the authors' conclusions.

Here are a few instances of the authors stating that they have shown that Omicron has higher transmissibility than Delta.

Line 61 (abstract): "Our statistical modelling suggests that Omicron is more transmissible than Delta in several countries including South Africa"

Line 71 (abstract): "Our data suggest... that Omicron has evolved increased transmissibility" This statement is particularly worrisome.

Line 99: 'One is that Omicron seems to be more transmissible than Delta' Re-state to something like: 'One is that Omicron seems to be spreading rapidly, especially relative to Delta's rate of spread.'

Line 123: 'Notably, although Delta is more transmissible than Alpha and Beta, our statistical analysis that the growth rate per generation of Omicron in South Africa was 3.31-fold higher than that of Delta': although this sentence does not explicitly state that Omicron has higher transmissibility than Delta, it could be easily interpreted as such given the first part of the sentence. I therefore suggest revising to:

Notably, our statistical analysis that the growth rate per generation of Omicron in South Africa was 3.31-fold higher than that of Delta.'

Line 132: 'and will outcompete' -> I suggest saying 'and may outcompete'. Whether Omicron will replace Delta depends presumably most on the extent of cross-immunity between them. If someone who was infected with Omicron is still susceptible to infection with Delta, then replacement of Delta by Omicron will be unlikely.

Line 320: 'our data suggest that Omicron has evolved increased transmissibility and attenuated pathogenicity' -> which data are suggesting increased transmissibility? Is this the relative growth rate analysis? If so, please restate for greater accuracy.

Line 324: 'higher transmissibility of Omicron' -> again, re-state, for example to: 'rapid rate of spread of Omicron'

Line 362: 'as shown in this study and others²², Omicron's transmissibility is higher than that of Delta,' Again, is this based off of the epidemiological analysis? If so, the conclusions are misstated because that analysis does not show this – it shows a higher growth rate (which can be explained by higher intrinsic transmissibility of Omicron OR a number of other processes, e.g., extent of immune escape).

Other concerns:

Figure 1 presents relative growth rates per generation. Growth rates are generally presented in calendar time, not viral generation time. Relative growth rates, calculated as the ratio of (e.g., daily) growth rates, would have their calendar time units cancel. (If the viral generation time is assumed to be the same for the variants considered, then, similarly, viral generation times would cancel, as is the case here in this manuscript.) So, in either case, presenting as 'Relative growth rate' rather than 'Relative growth rate per..' is more appropriate. If viral generation times were assumed to be different for different variants, you couldn't simply take the ratio, you would have to switch the units to calendar time first, and then take the ratio. This is all just to say that the relative growth rates presented should be presented as 'relative growth rate' rather than 'relative growth rate per...'

For reproducibility, the code for inferring relative growth rates should be made available on GitHub prior to the publication of this paper. (That is, the Bayesian statistical modeling code.)

Author rebuttals to First Revision:

Comments from reviewer 3:

I think the revisions to this manuscript have improved the presented work, but I have several remaining concerns.

Our reply:

We are happy to hear *“this manuscript have improved the presented work”*. In the revised manuscript, we fixed the remaining concerns raised by the reviewer. Please find the revised manuscript as well as our point-by-point replies below.

Conclusion that Omicron has higher transmissibility than Delta:

Reviewer 1 and reviewer 3 (me) both commented on the inappropriate conclusion by the authors, based on their analyses, that Omicron is “more transmissible” than Delta. The authors also state in their response letter (here in response to reviewer 1) that ‘We agree that the term ‘transmissibility’ is not appropriate to interpret the results of our statistical modelling analysis’. Despite this agreement, the manuscript still repeatedly states this conclusion, which is not explicitly supported by their analyses. The analyses indicate that Omicron is spreading more rapidly than Delta (that is, currently has a higher growth rate). But this could be because of Omicron’s ability to escape immunity, not intrinsically higher transmissibility of Omicron relative to Delta. To me, the remaining text concluding higher transmissibility is a big concern, as a paper published in Nature has the potential to be heavily cited, and this conclusion is not supported by their analyses. Almost certainly, this would prompt letters to Nature expressing disagreement with the authors’ conclusions.

Here are a few instances of the authors stating that they have shown that Omicron has higher transmissibility than Delta.

Line 61 (abstract): “Our statistical modelling suggests that Omicron is more transmissible than Delta in several countries including South Africa”

Line 71 (abstract): “Our data suggest... that Omicron has evolved increased transmissibility” This statement is particularly worrisome.

Our reply:

According to the suggestion, we modified the revised manuscript as below:

Line 61 (abstract): “Our statistical modelling suggests that Omicron is more transmissible than Delta in several countries including South Africa”

“Our statistical modelling suggests that Omicron **spreads more rapidly** than Delta...”

Line 71 (abstract): “Our data suggest... that Omicron has evolved increased transmissibility” This statement is particularly worrisome.

“... that Omicron has evolved **increased spread rate in the human population** and attenuated pathogenicity.”

Line 99: ‘One is that Omicron seems **to be more transmissible than** Delta’ Re-state to something like: One is that Omicron seems **to be spreading rapidly, especially relative to Delta’s rate of spread.**’

Our reply:

Fixed this sentence according to the suggestion (lines 97-98, page 4):

“One is that Omicron seems **to be spreading rapidly, especially relative to the spread rate of Delta, ...**”

Line 123: ‘Notably, **although Delta is more transmissible than Alpha and Beta, our statistical analysis that the growth rate per generation of Omicron in South Africa was 3.31-fold higher than that of Delta**’: although this sentence does not explicitly state that Omicron has higher transmissibility than Delta, it could be easily interpreted as such given the first part of the sentence. I therefore suggest revising to:

Notably, **our statistical analysis that the growth rate per generation of Omicron in South Africa was 3.31-fold higher than that of Delta.**’

Our reply:

Fixed this sentence according to the suggestion (lines 122-124, page 5):

“Notably, our statistical analysis showed the growth rate per generation of Omicron in South Africa was 3.31-fold higher than that of Delta.”

Line 132: ‘and will outcompete’ -> I suggest saying ‘and may outcompete’. Whether Omicron will replace Delta depends presumably most on the extent of cross-immunity between them. If someone who was infected with Omicron is still susceptible to infection with Delta, then replacement of Delta by Omicron will be unlikely.

Our reply:

According to the suggestion, we replaced “will” with “may” in the revised manuscript (line 131, page 5).

Line 320: ‘our data suggest that Omicron has evolved increased transmissibility and attenuated pathogenicity’ -> which data are suggesting increased transmissibility? Is this the relative growth rate analysis? If so, please restate for greater accuracy.

Our reply:

Similar to the Abstract (see above), we fix this sentence as below (lines 316-317, page 10):

“... our data suggest that Omicron has evolved increased spread rate in the human population and attenuated pathogenicity.”

Line 324: ‘higher transmissibility of Omicron’ -> again, re-state, for example to: ‘rapid rate of spread of Omicron’

Our reply:

According to the suggestion, we replaced “**higher transmissibility**” with “**rapid rate of spread**” in the revised manuscript (line 320, page 10).

Line 362: ‘as shown in this study and others²², Omicron’s transmissibility is higher than that of Delta,’ Again, is this based off of the epidemiological analysis? If so, the conclusions are misstated because that analysis does not show this – it shows a higher growth rate (which can be explained by higher intrinsic transmissibility of Omicron OR a number of other processes, e.g., extent of immune escape).

Our reply:

According to the suggestion, we rephrased “**Omicron’s transmissibility is higher than...**” with “**Omicron spreads more rapidly than...**” in the revised manuscript (line 358, page 11).

Other concerns:

Figure 1 presents relative growth rates per generation. Growth rates are generally presented in calendar time, not viral generation time. Relative growth rates, calculated as the ratio of (e.g., daily) growth rates, would have their calendar time units cancel. (If the viral generation time is assumed to be the same for the variants considered, then, similarly, viral generation times would cancel, as is the case here in this manuscript.) So, in either case, presenting as ‘Relative growth rate’ rather than ‘Relative growth rate per..’ is more appropriate. If viral generation times were assumed to be different for different variants, you couldn’t simply take the ratio, you would have to switch the units to calendar time first, and then take the ratio. This is all just to say that the relative growth rates presented should be presented as ‘relative growth rate’ rather than ‘relative growth rate per...’

Our reply:

Thank you for sharing your concerns about our statistical model. However, we are afraid that the average generation time [γ (gamma)] is not cancelled in the calculation of the relative growth rate (r) in our model even if the γ value is common among viral lineages. The reason is explained below:

In our model, the relative growth rate (r) is defined as:

$$r = \exp(\gamma/wb_1) = (\exp(b_1/w))^Y$$

where w is the time bin size. Therefore, the effect of γ on r is exponential and so is not cancelled in a linear scale. In other words, if we use a different fixed value of γ , the ratio between r_{Omicron} and r_{Delta} will change. Therefore, previous studies that used the statistical models similar to ours are cited in the revised manuscript [references 19 and 20: Vöhringer et al., *Nature*, 2021 (PMID: 34649268) and Obermeyer et al., *medRxiv*, 2021 (<https://doi.org/10.1101/2021.09.07.21263228>)]. **[Redacted]** in these previous studies, the “relative growth rate per viral generation time” was calculated and compared the spread rates among viral lineages.

Fig. R1. Estimation of the relative growth rate per viral generation time in previous studies. The relevant parts are annotated in red.

[Redacted]

According these reasons, we did not modify our statistical model in the revised manuscript.

However, to clarify that we assume $\gamma = 5.5$ day and that the estimated growth rate is the value per 5.5 day in the present study, we used the expression “relative growth rate per 5.5 day” instead of “relative growth rate per generation” in the y-axes of **Fig. 1a and 1c** in the revised manuscript. Moreover, we modified the relevant parts of the main text in the revised manuscript (e.g., line 121, page 5;).

For reproducibility, the code for inferring relative growth rates should be made available on GitHub prior to the publication of this paper. (That is, the Bayesian statistical modeling code.)

Our reply:

We archived the GitHub repository sharing the computational codes to estimate the relative growth rate of each SARS-CoV-2 lineage and cited this repository in the “Code availability” section (lines 1031-1035, page 29) and the reference list (reference 50; lines 1071-1072, page 30) in the revised manuscript:

The Sato Lab. Estimation of the relative growth rate of each viral lineage. Zenodo. doi: 10.5281/zenodo.5875972 (2022).

Comments from reviewer 3:

Re. Figure 1 – I still don't understand. If car A goes 60 miles per hour and car B goes 30 miles per hour (both rates), car A is twice as fast as car B. Not twice as fast per hour. The Obermeyer figure and legend they include shows a ratio of reproductive numbers on the y-axis. But the figure legend caption says 'growth rate versus date of lineage emergence'. A ratio of reproductive numbers is not a growth rate (reproductive numbers are dimensionless and a ratio of them does not give you a rate, e.g. per day growth rate or per viral generation growth rate). This also has not gone through peer review. Re. the Vohringer paper, I see that they do the same thing as the authors here (Sato et al.), but it still makes no sense to me that the relative growth rate of omicron vs delta depends on the generation time assumed. Car A can go 60 miles per hour, or alternatively 120 miles per 2 hrs or 120 miles per my class period, and car B can go 30 miles per hour or 60 miles per 2 hrs or 60 miles per my class period, but car A in any case is going twice as fast as car B. Car A wouldn't be going twice as fast if I register the cars' speeds in per hour units and only 1.5 times faster if I register the cars' speeds in per class period duration.

Our reply:

The reviewer mentioned that the term "*rate*" should mean a common difference in an arithmetic progression (e.g., car speed [km/hour] in the reviewer's example; **Fig. R1, left**). As the reviewer mentioned, the time unit (per hour) can be canceled when we calculate the ratio of the speeds of two cars. On the other hand, the value "r", which was referred to as a relative growth "*rate*" per generation in our study, corresponds to a common ratio in a geometric progression (**Fig. R1, right**).

[Redacted]

Fig. R1. Difference of the meaning of "r (*rate*)".

We now agree that the use of the term "*rate*" might be a bit confusing. Therefore, in the revised manuscript, we referred to "r" as a relative "effective reproduction number", instead of a relative "growth rate" per generation.

Regarding the terminology of "r" in our study, we discussed with Professor Akira Sasaki:

https://scholar.google.co.uk/citations?user=k_x9SIQAAAAJ&hl=en&oi=ao

Dr. Sasaki is an expert of mathematical modelling on virology and has recently published a paper at Nat. Ecol. Evol.:

<https://www.nature.com/articles/s41559-021-01603-z>

Dr. Sasaki agreed that the use of the term relative "effective reproduction number" to represent "r" in our statistical model is accurate and acceptable.